# Single-Cell Sequencing Reveals Novel Tumor Populations and Their Interplay with the Immune Microenvironment in a Pleomorphic Rhabdomyosarcoma

**DOI:** 10.3390/ijms262311420

**Published:** 2025-11-26

**Authors:** Elena E. Kopantseva, Alexander V. Ikonnikov, Maxim E. Menyailo, Timur I. Fetisov, Anastasia A. Korobeynikova, Kirill I. Kirsanov, Anastasia A. Tararykova, Beniamin Yu. Bokhyan, Nikolay A. Kozlov, Marianna G. Yakubovskaya, Evgeny V. Denisov

**Affiliations:** 1Research Institute of Molecular and Cellular Medicine, Peoples’ Friendship University of Russia (RUDN University), 115093 Moscow, Russia; y.kopantseva@gmail.com (E.E.K.); alex.v.ikonnikov@gmail.com (A.V.I.); max1989me@gmail.com (M.E.M.); timkatryam@yandex.ru (T.I.F.); shegolmay@gmail.com (A.A.K.); kkirsanov85@yandex.ru (K.I.K.); anastasiatararykova@gmail.com (A.A.T.); newbox13@mail.ru (N.A.K.); mgyakubovskaya@mail.ru (M.G.Y.); 2N.N. Blokhin National Medical Research Center of Oncology, 115522 Moscow, Russia; beniamin-bokhyan@mail.ru

**Keywords:** pleomorphic rhabdomyosarcoma, soft-tissue sarcoma, single-cell RNA sequencing, bioinformatics, immunotherapy

## Abstract

Pleomorphic rhabdomyosarcoma is a rare soft-tissue tumor that occupies an uncertain middle ground between rhabdomyosarcoma and undifferentiated pleomorphic sarcoma. With its relative rarity, aggressiveness, and lack of detailed characterization, it presents a challenging task for therapeutic treatment. In this case study, we use single-cell transcriptomics to investigate the heterogeneous landscape of pRMS and the tumor microenvironment. We demonstrate that the tumor populations in pRMS have a clear division into myogenic and non-myogenic clusters, with the non-myogenic clusters having more numerous communication links with the immune populations. All pRMS tumor clusters use the MIF-CD74 pathway to suppress the immune response, while APP, PTN, and CXCL12 signaling are employed predominantly by the non-myogenic tumor clusters. The cytotoxic T cells in pRMS bear markers of exhaustion (*LAG3*, *HAVCR2*, *EOMES*), and the macrophages express myeloid checkpoint-related genes (*SIGLEC1*, *SIRPA*, *CSF1R*, *HAVCR2*). This transcriptomic data suggests that targeting MIF and APP signaling in pRMS may have therapeutic potential; however, studies on multiple-patient cohorts, protein verification, and in vitro and in vivo validation are still needed for clinical actionability.

## 1. Introduction

Pleomorphic rhabdomyosarcoma (pRMS) is a rare, extremely aggressive, lesser-studied subtype of rhabdomyosarcoma (RMS), displaying both pleomorphic and skeletal muscle features [1,2]. pRMS mostly occurs in adults [3], and the most common location for the tumor is the deep soft tissues of the lower limbs [4]. The prognosis for pRMS is dismal: local recurrences and distant metastases are common; 80% of patients succumb to the disease, with the median survival period at the metastatic stage being 7.3 months [1,4]. Differential diagnoses of dedifferentiated liposarcoma and high-grade undifferentiated pleomorphic sarcoma prolong the wait for the correct treatment [2]. The tumor response to chemotherapy remains poor and short-lived, compared to other soft-tissue sarcomas [5,6]. Surgical resection with pre- or postoperative radiotherapy remains the standard protocol for localized treatment of pRMS [6]. There is evidence that immunotherapy holds potential for pRMS, as demonstrated in the clinical case of a PD-L1-positive pRMS patient [7]. However, not all pRMSs are PD-L1-positive [8]. In some cases of pRMS, addition of immunotherapy to standard chemotherapy is suggested to be effective [9]. Therefore, there is an urgent need to acquire more knowledge on the mechanisms of pRMS and molecular features of tumor and immune cell populations.

Genomic analysis of pRMS showed genomic alterations similar to those of pleomorphic sarcomas with a high degree of genomic instability (duplication, chromothripsis, and kataegis), placing pRMS further away from other types of RMS [10,11]. pRMS tumor cells have genomic alterations in *TP53*, *RB1*, *CDKN2A*, *ARID1B*, *NOTCH3*, *CDKN2B*, *CDK4*, *KRAS*, *FGFR1*, *NF1*, *PTEN*, *AKT3*, *TERT*, and other genes [11,12]. An association with a genetic predisposition toward a DNA mismatch repair deficiency, also known as Lynch Syndrome, has also been reported for pRMS [13].

Bulk RNA expression profiling of pRMS demonstrated an upregulation of genes involved in cell cycle regulation, epigenetic modification, resistance to apoptosis, cancer cell metabolism, RTK/MAPK, AKT/PIK3CA/mTOR, Wnt, Hedgehog, and JAK/STAT pathways [12]. The expression of *TGFB1* is upregulated in pRMS, in comparison with the healthy muscle. pRMSs with high immune infiltrate levels are enriched in the cytokine–cytokine, antigen presentation, JAK-STAT, and TCR signaling pathways. High-immune-infiltrative pRMS also shows an upregulation of the *LAG3*, *IDO1*, and *IFI30* immunosuppressive markers [10].

Multiplex immunofluorescent staining led to a hypothesis that a subset of pRMSs contains quiescent lymphoid cells that are affected by the muscle microenvironment and the expression of the dystrophin gene [10]. Single-cell cytometric protein analysis of the mouse model of pRMS revealed an elevated expression of adipogenic markers and heterogeneous highly exocytic tumor populations, which likely promote the transformation of the tumor microenvironment (TME) towards the pro-fibrotic type [14]. Despite these advancements, the knowledge of pRMS biology remains incomplete and requires additional research.

Single-cell RNA sequencing (scRNA-seq) is a powerful technology that allows simultaneous characterization of cell types, cell cycle stages, tumor cell phylogeny, metabolic pathways, cell–cell interactions, cell trajectories, gene regulatory networks, and other molecular features at the single-cell level. In this study, we investigated the single-cell transcriptional landscape of pRMS in a 62-year-old patient for the first time focusing on the differentially expressed genes, enriched biological processes in pRMS tumor and TME cell populations, cell–cell interactions between the tumor and immune cells, and potential therapeutic hypothesis-level implications.

## 2. Results

### 2.1. pRMS Case Presentation

A 62-year-old female patient discovered a tumor in the soft tissues of her right thigh in November 2019. After a core biopsy of the neoplasm, the diagnosis of undifferentiated sarcoma was made. In 2020, five courses of combined doxorubicin (60 mg/m^2^) and ifosfamide (7.5 mg/m^2^) and one course of combined doxorubicin (90 mg/m^2^) and dacarbazine (900 mg/m^2^) chemotherapy were carried out pre surgery, with the resulting effect of stabilization according to RECIST 1.0. Afterwards, distant radiotherapy (single dose—3.0 Gr; total dose—44 Gr) was carried out, followed by the wide excision of the neoplasm in the right thigh.

After 3 months, the structure of the thoracic and lumbar vertebrae showed the presence of foci without fixed borders and sclerotic foci, which accumulated the radiopharmaceutical drug. According to positron emission tomography and computed tomography (PET/CT) scans, by February 2023, the soft tissues of the patient’s right upper thigh contained a formation with bumpy contours. No other centers of radiopharmaceutical drug accumulation were noted. Magnetic resonance imaging (MRI) conducted in May 2023 revealed that the soft tissues of the anterior–inferior surface of the upper third of the right thigh contained a heterogeneous solid tumor with an irregular contour. Taking into consideration the pain syndrome and the risk of tumor disintegration, surgical resection of the soft tumor tissue in the right thigh was performed in May 2023.

The post-surgical histological analysis of the removed tumor in June 2023 showed that the tumor is represented by fields of atypical explicitly polymorphic and spindle-like cells. To clarify the diagnosis, immunohistochemical analysis was performed. The results revealed the presence of desmin (DES) in the majority of tumor cells and the focal expression of myogenin (MYOG) and MYOD1 (Figure 1A). The traits revealed through immunohistochemistry suggested the morphological picture of G3 pRMS. Two months after the surgery, the patient succumbed to the progressing disease.

### 2.2. Annotation of pRMS Cell Clusters

In total, after filtering, 3099 cells were identified, with a median 2826 genes per cell. A total of 14,955 genes were detected in the sample, and the median UMI count was 5221 per cell. The percentage of confidently mapped reads in the analyzed cells was 91.87%. The percentage of total filtered cells was 12.84% (456/3555 cells). The duplicate rate was 2.7% (96/3555 cells). For this dataset, SoupX was not used to remove RNA contamination, since the data did not contain elevated values of contaminating RNAs (Section 4.6 in the Methods contains the full ambient-RNA rationale explanation). The scRNA-seq data presented in the study is openly available in Zenodo under the accession number 17349024 (Data Availability).

Cell clustering using the Leiden algorithm and uniform manifold approximation and projection (UMAP) analysis led to the identification of 15 cell clusters. To define the cell subsets, we analyzed their cell ploidy and compared their gene profiles against the canonical set of markers for each cell type (Figure 1C; Appendix A, Sheet 1, markers for cell annotation; [15,16]). The final annotated clusters were defined as follows: tumor cells (1475 cells), proliferating tumor cells (380 cells), mesenchymal stem-like cells (178 cells), fibroblasts (249 cells), endothelial cells (69 cells), vascular smooth muscle cells (VSMCs) (78 cells), T cells (344 cells), and macrophages (326 cells; Figure 1B; Appendix A, Sheet 3, cell counts). After the initial annotation, each cell type was taken for further characterization.

### 2.3. pRMS Tumor Cell Heterogeneity

#### 2.3.1. Cell Ploidy and Differential Gene Expression

Cell ploidy analysis was used to identify tumor clusters (TCs) in the scRNA-seq dataset. Clusters 4–5, 7–8, 10–11, and 13–15 contained a low percentage (<30%) of aneuploid cells and were annotated as normal, considering evidence of low aneuploidy levels in normal cells [17]. Cluster 1 contained 98.9% (533/539 cells) of aneuploid cells; cluster 2—60.6% (257/424 cells); cluster 3—92.9% (352/379 cells); cluster 6—73.8% (203/275 cells); cluster 9—96.8% (121/125 cells); and cluster 12—100% (83/83 cells). Clusters 1–3, 6, 9, and 12 were assigned as tumor clusters (Figure 2A,B; Appendix A, Sheet 2, validation of tumor vs. TME). The annotation characterizing cell ploidy in pRMS populations is visualized as a UMAP plot in Figure 2C.

Gene expression levels were compared between six tumor and TME cell clusters (Log2 Fold Change > 1, pct. > 0.5, adj. *p*-value < 0.05; Figure 2D). In addition, over 70% (1277/1825 cells; Log2 Fold Change > 1, pct. > 0.7, adj. *p*-value < 0.05) of tumor cells were found to be functionally enriched in genes associated with endodermal cell differentiation, regulation of cell migration, extracellular matrix (ECM) organization, regulation of cell differentiation, encapsulating structure organization, and the Wnt signaling pathway (Figure 2E). The genes with an elevated expression in TCs are related to the ECM (*COL1A1*, *LAMA5*), growth factors (*MFGE8*, *PTN*), the formation of lipid rafts (*GPC1*, *SORT1*, *PGRMC2*, *IGF1R*, *CD24*), leukocyte transmigration (*CD99L2*, *EDIL3*), chemoresistance (*RUNX2*), and interaction with the immune cells (*CD24*, *CDH2*) (Figure 2E).

#### 2.3.2. Functional Characterization

Reclustering of aneuploid cells using the Leiden algorithm was performed, and eight TCs were identified (Log2 Fold Change > 1, pct. > 0.5, adj. *p*-value < 0.05): RSPO3^+^ (40 cells), fibroblast-like (287 cells), proliferating (337 cells), S-phase (135 cells), proliferating myoblast-like (21 cells), myocyte-like (191 cells), MEST^+^ (339 cells), and LENG8^+^ TCs (241 cells; Figure 2F,G; Appendix A, Sheet 1, pRMS tumor clusters; Appendix A, Sheet 3, cell counts).

The RSPO3^+^ TC is enriched (Log2 Fold Change > 1, pct. > 0.5, adj. *p*-value < 0.05) in genes related to the ECM (*ADAMTS7*, *ADAMTS14*, *FLRT2*, *COL24A1*, *MMP2*, *HMCN1*), regulation of cell migration (*RAP2B*, *CSF1*, *MDK*, *DPYSL3*, *SULF1*, *ROBO1*, *AMOT*), and Wnt (*FZD1*, *FZD4*, *DAAM2*, *LEF1*, *RSPO3*, *NKD2*) and Hippo (*FZD1*, *FZD4*, *LEF1*, *DCHS1*, *TEAD2*, *AMOT*, *NKD2*) signaling pathways. Considering these data, the *RSPO3*^+^ TC may be linked to the migratory phenotype.

The fibroblast-like TC has high expression levels (Log2 Fold Change > 1, pct. > 0.5, adj. *p*-value < 0.05) of genes encoding collagens and other ECM proteins (*COL3A1*, *COL6A3*, *COL12A1*, *LUM*, *MXRA5*, *MXRA8*), matrix-remodeling enzymes (*MMP2*, *MMP14*, *ANPEP*), and immune cell migration factors (*COL6A3*, *SPON2*, *CSF1*). This cluster may have a role in communication with immune cells and facilitation of tumor cell migration through ECM remodeling, as shown previously for other tumors [18].

The proliferating TC is enriched (Log2 Fold Change > 1, pct. > 0.5, adj. *p*-value < 0.05) in genes related to proliferation (*MKI67*, *TOP2A*, *BUB1*), chromatid segregation (*SPAG5*, *KIF14*, *NCAPG*, *KIF23*, *KIF11*, *KIF22*), spindle organization (*NDC80*, *CENPE*, *TPX2*, *KIFC1*, *PRC1*), and cytokinesis (*CIT*, *ANLN*, *ESPL1*, *KIF4A*, *NUSAP1*).

The S-phase TC is enriched (Log2 Fold Change > 1, pct. > 0.5, adj. *p*-value < 0.05) in genes related to DNA replication (*POLA1*, *POLD1*, *POLE2*, *MCM2/3/4/5/6/7*) and DNA repair (*SLF1*, *LIG1*, *XRCC3*, *FANCA*, *UNG*, *WDR76*, *FANCG*).

The proliferating myoblast-like TC and the myocyte-like TC are the only pRMS TCs with upregulated myogenic signatures (Log2 Fold Change > 1, pct. > 0.5, adj. *p*-value < 0.05). The myocyte-like TC expresses master regulators of myogenesis and muscle repair (*MYOD1*, *MYOG*, *MEF2C*) and cytoskeletal proteins typical of mature myocytes (*DES*, *TNNI1*, *TNNI2*, *TNNT1*, *TNNT3*, *TNNC1*, *TNNC2*, *MYL1*, *MYL4*, *MYL6B*, *MYLPF*, *MYOM3*, *DYSF*, *MYMX*). Interestingly, this cluster also has an elevated expression of *MYH3*, an embryonic form of myosin heavy chain, several cardiac cytoskeletal genes (*ACTC1*, *TNNT2*), and *SERPINB9*, which has previously been shown to protect tumor cells from granzyme B secreted by cytotoxic T lymphocytes [19]. The proliferating myoblast-like TC only expresses a fraction of the myogenic cytoskeletal genes (*TNNI1*, *TNNT3*, *MYL1*, *ACTC1*) and has additional enrichment in proliferation genes (*MKI67*, *BUB1*, *TOP2A*). This places cells in this TC closer to immature myoblasts, which can still undergo division, than to mature myocytes. However, the proliferating myoblast-like TC lacks the expression of canonic myogenic transcription factors (*MYOG*, *MYOD1*), which are present in normal myoblasts.

The MEST^+^ TC and LENG8^+^ TC express insufficient differentially expressed genes (DEGs) (Log2 Fold Change > 1, pct. > 0.5, adj. *p*-value < 0.05) to receive functional characterization.

### 2.4. pRMS Tumor Microenvironment

#### 2.4.1. T Lymphocytes

The T-cell cluster displays upregulated expression (Log2 Fold Change > 1, pct. > 0.5, adj. *p*-value < 0.05) of classical T-cell (*CD2*, *CD3E*, *CD8A*, *TRAC*, *TRBC2*), interleukin and cytokine (*IL32*, *IL2RB*, *IL2RG*, *CCL5*, *CXCR4*), GPCR-coupled signaling (*ADGRE5*, *ARHGAP4/9/15*, *ARHGEF1*, *ARHGDIB*), stem (*CD44*), and exhaustion (*TIGIT*) marker genes. To investigate the diversity of T lymphocytes, we performed their reclustering using the Leiden algorithm. Five cell clusters were identified (Log2 Fold Change > 1, pct. > 0.5, adj. *p*-value < 0.05): CD4^+^ T cells (16 cells), CD4^+^ regulatory T cells (T reg cells; 69 cells), proliferating CD8^+^ T cells (40 cells), cytotoxic CD8^+^ T cells (131 cells), and IL7R^+^ T cells with mixed CD4^+^ and CD8^+^ markers (88 cells; Figure 3B; Appendix A, Sheet 3, pRMS T-cell clusters; Appendix A, Sheet 3, cell counts).

Most CD4^+^ T cells in pRMS possess markers (Log2 Fold Change > 1, pct. > 0.5, adj. *p*-value < 0.05) of T reg cells (*CD4*, *FOXP3*, *TNFRSF4*, *MAF*, *IKZF2*, *TBC1D4*; Appendix A, Sheet 2, TME gene signatures) [20]. The elevated expression of genes of the TNF superfamily (*TNFRSF18*, *TNFRSF1B*, *TNFAIP3*, *LTB*) and markers of effector T reg cells (*ICOS*, *CTLA*, *TIGIT*) in the CD4^+^ T reg cell cluster suggests their increased proliferation and anti-inflammatory response, as seen in previous studies [21,22]. The CD4^+^ T reg cells also contain a high expression of *LAYN*, which has been previously linked to immunosuppressive functions of T reg cells in skeletal undifferentiated pleomorphic sarcoma [16].

The proliferating CD8^+^ T cells mostly express (Log2 Fold Change > 1, pct. > 0.5, adj. *p*-value < 0.05) markers of cell cycle (*MKI67*, *TOP2A*, *MCM3/5/6/7*, *MYBL2*) and chromatin reorganization (*EZH2*, *LMNB1*, *LMNB2*, *UHRF1*). The cytotoxic CD8^+^ T-cell cluster (*CD8A*, *KLRK1*, *GZMA*, *GZMH*, *NKG7*, *CTSW*) harbors an elevated expression (Log2 Fold Change > 1, pct. > 0.5, adj. *p*-value < 0.05) of *GZMK*, which is hypothesized to be a characteristic of the CD8^+^ subgroup enriched in stem-like and memory qualities [23], inflammatory cytokines and their receptors (*CCL4*, *CCL5*, *CCR5*), and markers of the exhausted T-cell state (*LAG3*, *HAVCR2*, *EOMES*; Appendix A; Sheet 2, TME gene signatures) [20,24]. The CD8^+^ T-cell exhaustion signature (*LAG3*, *HAVCR2*, *EOMES*, *PDCD1*) is expressed (Log2 Fold Change > 1, pct. > 0.5, adj. *p*-value < 0.05) exclusively in the cytotoxic CD8^+^ T-cell cluster among the five T-cell clusters (Figure 3D).

IL7R^+^ T cells harbor markers (Log2 Fold Change > 1, pct. > 0.5, adj. *p*-value < 0.05) previously reported as promoting a naive, stem-like phenotype in T cells (*IL7R*, *TCF7*, *ZFP36L2*, *TXNIP*) [20,25,26,27]. The IL7R^+^ T cells do not express the CD8^+^ T-cell exhaustion signature that was demonstrated for the cytotoxic CD8^+^ T-cell cluster.

#### 2.4.2. Macrophages

pRMS macrophages exhibit (Log2 Fold Change > 1, pct. > 0.5, adj. *p*-value < 0.05) key genes (*CD163*, *APOE*, *CTSZ*, *CTSB*, *MAF*, *GLUL*) from the previously described signature of the anti-inflammatory cysteine cathepsin-positive tumor-associated macrophages (Appendix A, Sheet 2, TME gene signatures) [28]. This cluster also has immunosuppressive (*IL10RA*, *CD209*, *MAF*, *GRN*), myeloid and immune checkpoint-related (*SIGLEC1*, *SIRPA*, *CSF1R*, *HAVCR2*; [29,30]; Figure 3E), stem cell (*CD44*), cytoskeletal regulator (*SDC*), and anti-tumor effector (*CD14*, *CD40*, *TLR2*) markers, as well as various receptor genes (*C3AR1*, *ADGRE5*, *PECAM1*, *ITGB2*). The TOP GO processes enriched in this cluster (Log2 Fold Change > 1, pct. > 0.5, adj. *p*-value < 0.05) are related to receptor-mediated endocytosis, the inflammatory response, regulation of cytokine and interleukin-6 production, phagocytosis, endocytosis, the reactive oxygen species metabolic process, and canonical Nf-kB signal transduction.

#### 2.4.3. Fibroblasts

pRMS fibroblasts are marked by an elevated expression (Log2 Fold Change > 1, pct. > 0.5, adj. *p*-value < 0.05) of myofibroblastic cancer-associated fibroblast signature genes (*POSTN*, *TAGLN*, *THY1*, *COL1A1*, *COL1A2*, *LOXL2*, *MMP11*, *FAP*, *PDGFRB*, *FBLN1*, *LRRC15*, *TNC*, *THBS2*) [31] and genes related to cell adhesion and ligand–receptor interactions (*ITGAV*, *FN1*, *C3*, *F2RL2*). Additional reclustering (Log2 Fold Change > 1, pct. > 0.5, adj. *p*-value < 0.05) yielded *COL6A*^+^ fibroblast and *ARGLU*^+^ fibroblast cell clusters (Figure 3C; Appendix A, Sheet 4, pRMS fibroblast clusters).

COL6A^+^ fibroblasts have a particularly robust elevation in expression (Log2 Fold Change > 1, pct. > 0.5, adj. *p*-value < 0.05) of collagens and metalloproteases (*COL3A1*, *COL6A1*, *COL6A2*, *COL6A3*, *COL10A1*, *MMP2*, *MMP11*). The top pathways enriched in this cluster (Log2 Fold Change > 1, pct. > 0.5, adj. *p*-value < 0.05) are related to ECM remodeling, cell motility, Golgi transport, and the protein catabolic process. The DEGs of this cluster most closely resemble the gene signatures of myofibroblastic cancer-associated fibroblasts [31] due to the abundance of genes related to matrix deposition.

ARGLU^+^ fibroblasts exhibit a high expression (Log2 Fold Change > 1, pct. > 0.5, adj. *p*-value < 0.05) of RNA-binding proteins (*ARGLU1*, *TIA1*, *CELF1*, *RBM25*, *CIRBP*) and mRNA splicing regulators (*RBM6*, *CCNL2*, *DDX5*, *DDX17*). They are functionally enriched in processes related to mRNA processing, metabolic processing, and splicing.

#### 2.4.4. Mesenchymal Stem-like Cells

Mesenchymal stem-like cells contain (Log2 Fold Change > 1, pct. > 0.5, adj. *p*-value < 0.05) markers previously shown to be regulators of cancer cell stemness (SOX4, RAB13, EZH2) [32,33,34]. This cluster is located adjacent to the tumor cell clusters on the initial UMAP visualization and has 25.85% (38/147 cells) of aneuploid cells.

#### 2.4.5. Endothelial Cells and VSMCs

The endothelial cells in pRMS are enriched (Log2 Fold Change > 1, pct. > 0.5, adj. *p*-value < 0.05) in processes related to cell migration. The elevated expression of leukocyte transendothelial migration-related genes (*CTNND1*, *PXN*, *PIK3R3*, *PTK2*, *GNAI2*, *CDH5*, *CLDN5*, *PECAM1*, *CTNNB1*, *RAPGEF3*, *ICAM2*) may suggest the recruitment of monocytes, granulocytes, and lymphocytes to the tumor, considering previous data [35]. Other upregulated genes in this cluster include *NECTIN2* and *APP*, which are involved in cell adhesion and in encoding the amyloid precursor protein.

The VSMCs display elevated expression (Log2 Fold Change > 1, pct. > 0.5, adj. *p*-value < 0.05) of mesenchymal stem cell-like (*THY1*, *CD248*, *PDGFRB*), cell adhesion (*ITGAV*, *ITGB1*), and fibroblast marker genes (*COL1A2*, *COL3A1*, *COL5A2*, *COL6A1*, *COL6A2*).

### 2.5. Tumor–Immune Cell Communications in pRMS

Based on the transcriptomic data, we used CellChat to predict the potential ligand–receptor pairs in pRMS cell clusters, taking into account the gene expression markers previously uncovered through DEG analysis. Considering the signs of exhaustion in T-cell clusters and the myeloid checkpoint-related genes in macrophage clusters (Log2 Fold Change > 1, pct. > 0.5, adj. *p*-value < 0.05), we focused on the effect that pRMS tumor and immune cell clusters could potentially exert on each other. The MEST^+^ TC and LENG8^+^ TC were merged with the fibroblast-like TC for this analysis due to not having sufficient DEGs to conduct characterization of ligand–receptor pairs.

According to the overall number and strength of interactions (Figure 4A,B), non-myogenic (RSPO3^+^, fibroblast-like, proliferating, and S-phase) TCs have the potential to send stronger and more numerous signals to immune cells than myogenic (proliferating myoblast-like and myocyte-like) TCs (Log2 Fold Change > 0, pct. > 0.1, adj. *p*-value < 0.05). In return, the immune clusters likely send more signals to the S-phase TC than to any other TC (Log2 Fold Change > 0, pct. > 0.1, adj. *p*-value < 0.05). This suggests that pRMS TCs may interact with immune cells in different ways.

All TCs likely use the MIF–CD74 axis to target immune cells, with the RSPO3^+^ TC, proliferating myoblast-like TC, and myocyte-like TC being the most likely to initiate this interaction (Log2 Fold Change > 0, pct. > 0.1, adj. *p*-value < 0.05; Figure 4C). The MIF–CD74 pathway has been previously shown to promote the pro-tumor phenotype in macrophages and decrease CD8^+^ T-cell infiltration in Ewing’s sarcoma [36]; hypothetically, the same process could be happening in pRMS.

Non-myogenic TCs demonstrate additional potential communication links with immune cells that are predominantly not observed or observed with lower probability for myogenic TCs: APP–CD74, PTN–SDC3, and CXCL12–CXCR4 (Log2 Fold Change > 0, pct. > 0.1, adj. *p*-value < 0.05; Figure 4C). APP signaling is observed between non-myogenic TCs, T cells, and macrophages. PTN signaling is used by non-myogenic TCs to target all immune cells, except proliferating CD8^+^ T cells. Finally, the CXCL12 ligand is likely used by non-myogenic TCs to target IL7R^+^ T cells, CD4^+^ T cells, proliferating CD8^+^ T cells, and cytotoxic CD8^+^ T cells. All three pathways have previously been shown to restrict immune cell influx, alter CD8^+^ T-cell activity, and promote T-reg-cell accumulation [37,38,39], which could suggest a similar role in pRMS tumor–immune communication.

Immune cells in pRMS have numerous potential communication links to TCs, with the most probable being GRN–SORT1, SEMA4D-PLXNB2, PTN–SDC3, APP–TNFRSF21, GZMA–PARD3, and SIGLEC1–SPN (Log2 Fold Change > 0, pct. > 0.1, adj. *p*-value < 0.05; Figure 4D). The GRN–SORT1 pathway, predominantly used by macrophages to signal to all TCs, has the highest communication probability among the listed pathways. Progranulin has been shown to cause dedifferentiation and increased proliferation of the cancer stem cell pool in a breast cancer study [40]. This opens up an interesting discussion on the role of pro-tumor macrophages in pRMS and their effect on TCs.

## 3. Discussion

pRMS is a rare, aggressive type of RMS that exhibits the presence of non-myogenic (pleomorphic) and myogenic tumor cell populations, thereby combining the qualities of RMS and undifferentiated pleomorphic sarcoma. This presents a challenge for pRMS therapeutic treatment. In this report, we present the first single-cell transcriptome analysis of tumor and TME cell populations in pRMS and their functional interactions.

pRMS contains myogenic and non-myogenic tumor cell populations. Proliferating myoblast-like and myocyte-like TCs in pRMS bear an expression pattern similar to the committed and differentiated myogenic populations previously described for aRMS and eRMS [41]. However, we did not identify a TC with a gene signature similar to muscle stem-like tumor cells, which has been identified for aRMS and eRMS [41]. According to GO analysis, the RSPO3^+^ TC has potential for cell migration, the fibroblast-like TC remodels the ECM, which can aid cell migration, and the proliferating, S-phase, and proliferating myoblast-like TCs contribute to tumor growth through cell proliferation. According to cell–cell interaction analysis, all TCs communicate with immune cells through the MIF–CD74 axis. Non-myogenic TCs have more robust communication ties with immune cells, which is an interesting phenomenon that deserves further investigation. Non-myogenic TCs have the potential to regulate the immune cells through APP–CD74, PTN–SDC3, and CXCL12–CXCR4 signaling.

T lymphocytes in pRMS include effector CD4^+^ T cells, CD4^+^ T reg cells, cytotoxic CD8^+^ T cells, proliferating CD8^+^ T cells, and IL7R^+^ T cells. Cytotoxic CD8^+^ T cells express granzyme genes (*GZMA*, *GZMH*, *GZMK*) and possibly send signals to the TCs through the GZMA–PARD3 axis. At the same time, cytotoxic CD8^+^ T cells display markers of exhaustion (*LAG3*, *HAVCR2*, *EOMES*), which is consistent with bulk transcriptomic studies of pRMS [10].

The pRMS macrophages likely play a pro-tumor role. Their expression of cysteine cathepsins likely helps remodel the ECM and promote tumor invasiveness, as shown in previous papers [42]. pRMS tumor cells affect the macrophages through SIGLEC1 and CSF1R receptors, possibly preventing tumor cell phagocytosis. The expression of GRN by pRMS macrophages might be linked to dedifferentiation and proliferation of tumor cells, as suggested by previous studies [40].

It has been reported previously that pRMS patients benefit from the PD-1 inhibitor nivolumab [7]. However, in our cell–cell communication analysis, we were not able to detect PDL1 signaling (CD274–PDCD1) between tumor and immune cells in pRMS, and the communication probability for PDL2 signaling (PDCD1LG2–PDCD1) in the aforementioned cells is shown to be minimal (Figure 4C). Different levels of PDL1 expression between pRMS cases have been reported previously [8], which could explain our PDL1 signaling results. Based on the transcription-level CellChat data, MIF and APP tumor–immune signaling has a high probability of taking place in pRMS. The cell–cell interaction analysis shows that MIF signaling is initiated by all pRMS TCs, while APP signaling has a strong probability of happening between non-myogenic clusters and immune cells. Experimental data on anti-MIF and anti-APP oncotherapy already exists, with vaccination targeting of APP being discussed as a therapy for several cancers [37] and anti-MIF treatment having potential for treatment of undifferentiated pleomorphic sarcoma [43]. Therefore, we offer a hypothesis that targeting MIF could affect the immunosuppressive MIF-CD74 interaction between all pRMS TCs and immune cells in pRMS. Targeting APP could provide an additional effect on non-myogenic TCs with their more robust ties to immune populations. This hypothesis is based on transcription-level data and is not clinically actionable without studies on larger cohorts, protein verification, and in vitro and in vivo experiments.

While the other samples of pRMS and archival tissue for this particular case of pRMS were unavailable to validate the obtained findings, we have found previous reports of the tumor–immune roles of the discussed pathways in other cases of sarcoma. A single-cell study of the mouse model of UPS and human sarcoma cell lines has shown the presence of the MIF-CD74 and APP-CD74 tumor–immune axis and the immunosuppressive effect of MIF-CD74 on an in vitro level [43]. As we discussed previously, undifferentiated pleomorphic sarcoma is closely linked to pRMS, and therefore, it is likely that they share similar signaling mechanisms.

This case report contains a number of limitations. We were able to examine only one case of pRMS with scRNA-seq due to the rarity of this malignancy, and further single-cell studies are needed. Single-cell transcriptomics provides detailed information on the cell populations at the mRNA level, but protein verification is still required. In addition, the study lacks information on the spatial relationships between cell populations in the pRMS ecosystem that can be obtained by different spatial omics technologies. The cell–cell interaction results especially need verification on a protein and spatial level. Finally, the obtained findings are not sufficient to elucidate the pRMS mechanisms in light of the tumor ecology concept recently proposed to highlight the complexity of malignancies [44].

Future directions include immunohistochemical confirmation of MIF-CD74, APP-CD74, and CXCL12–CXCR4 interactions between pRMS tumor and immune cells and testing of the MIF and APP inhibition effect on pRMS in vitro and in vivo. Overall, these experiments will provide evidence on whether the described tumor–immune interactions are useful as immunotherapy targets in pRMS therapy.

In conclusion, this study is the first case of scRNA-seq analysis of pRMS describing the tumor and TME cell populations and demonstrating the repressive interactions between tumor and immune cells on a transcriptional level. However, this is a single-patient transcriptomic case study without protein-level or spatial validation, and no clinical actionability is claimed.

## 4. Methods

### 4.1. Consent and Approval

Informed consent for the study and future publication was obtained from the patient. The study was conducted in accordance with the Declaration of Helsinki and approved by The Ethics Committee of N.N. Blokhin National Medical Research Center of Oncology (27 October 2020; approval number: 2020-42).

### 4.2. Histological Analysis

The pRMS tissue samples were fixed in formalin (Biovitrum, Saint-Petersburg, Russia) for 48 h. Dehydration and embedding in paraffin (Biovitrum, Russia) were performed following routine methods. For histopathological analysis, hematoxylin and eosin (H&E) staining (Biovitrum, Russia) was performed on formaldehyde-fixed, paraffin-embedded tissue samples. Microscopic analysis of three stained sections of the tissue was performed on an Olympus CX23 light microscope (Olympus, Tokyo, Japan).

### 4.3. Immunohistochemistry

Immunohistochemical analysis with a panel of antibodies was performed for each 4 μm formalin-fixed paraffin-embedded tissue section using standard techniques. The staining was performed using the fully automated diaminobenzidine antigen retrieval system (Benchmark ULTRA; Ventana Medical Systems, Tucson, AZ, USA), with appropriate controls. The following antibodies were used: rabbit monoclonal anti-myogenin (F5D: prediluted, Cell Marque, Rocklin, CA, USA) and mouse monoclonal anti-desmin (D33: 1:200, Dako Inc., Glostrup, Denmark). Antigen–antibody complexes were highlighted with brown chromogen (3,3-diaminobenzidine tetrahydrochloride, ultraView Universal DAB Detection Kit, Ventana Medical Systems). Quantification was performed by counting signal-positive cells in 6 to 10 high-powered fields (×40 magnification) with an Olympus CX23 light microscope (Olympus, Japan).

### 4.4. Sample Preparation for Single-Cell RNA Sequencing

A pRMS tumor sample of 20–25 mg was excised. The tumor sample was placed into the fixation buffer (10x Genomics, Pleasanton, CA, USA) immediately after excision. The fixation and dissociation of the sample was performed according to the Tissue Fixation and Dissociation Protocol for Chromium-Fixed RNA Profiling (10x Genomics, USA). The cells were counted using an acridine orange/propidium iodide stain buffer (Logos Bioscience, Anyang, Gyeonggi-do, Republic of Korea) on a LUNA-FL Dual Fluorescence Cell Counter (Logos Bioscience, Korea). The cell suspension was stored at −80 °C with 100 µL of Enhancer solution (10x Genomics, USA) and 275 µL of 50% glycerol. Before the Chromium-Fixed RNA Profiling experiment, the cells were thawed, centrifuged, resuspended in 0.5× PBS with 0.02% BSA, and counted again. In total, 8000 cells were used in the scRNA-seq experiment.

### 4.5. Single-Cell RNA Library Construction and Sequencing

The scRNA-seq libraries were made according to the Chromium-Fixed RNA Protocol (10x Genomics, USA). The GEM particles were generated on Chromium iX/X (10x Genomics, USA). The amplification steps were performed on the RT-PCR machine QuantGene 9600 (Bioer, Hangzhou, China). The scRNA-seq libraries were sequenced on the Genolab M platform (Genemind, Shenzhen, China) with the following program: 28 cycles for read 1 and 90 cycles for read 2.

### 4.6. Single-Cell RNA Sequencing Quality Control and Data Processing

Demultiplexing of the scRNA-seq data was conducted using the Cell Ranger 7.1.0 (10x Genomics, USA) pipeline. The data was aligned to the GRCh38-2020-A reference genome. Seurat package 5.0.3 [45] was used for bioinformatic analysis. The following parameters were included in the preprocessing Quality Control (QC) workflow: 500 < nFeature < 8000, 500 < nCount < 30,000, percent mt < 5%, and genes present in a minimum of three cells. The percentage of expressed genes that are considered to be widely known contaminants in single-cell RNA-seq samples was analyzed: mitochondrial genes (*MT-**, where * represents a specific gene identifier), ribosomal genes (*RPS-**, *RPL-**, where * represents a specific gene identifier), hemoglobin genes (*HBA*, *HBB*), and cell stress genes (*FOS*, *JUN*, *HSPA1A*). None of the genes in this group exhibited an absolute percentage over 1% (Appendix A, Sheet 5, contamination summary). A decision was made not to use SoupX to remove RNA contamination, since the data did not contain elevated values of contaminating RNAs, and avoiding the possible effects of expression matrix correction on data interpretation was preferable. The total percentage of filtered cells was 12.84% (456/3555 cells).

Doublet detection was performed using the DoubletCollection package [46]. Four additional packages were also used to identify doublets: ‘DoubletFinder’ [47], ‘cxds’ [48], ‘scDblFinder’ [49], and ‘Scrublet’ [50]. Cells identified by at least one of the methods were considered doublets. The duplicate rate was 2.7% (96/3555 cells).

The SCTransform with vst.flavor = ‘v2’ method applied to the expression matrices was performed to carry out data normalization.

Clustering/UMAP parameters (resolution, n.neighbors) for each figure are described in the analysis script pRMS analysis. R published in the Zenodo repository (Line 41–44,145–151,188–193; Data Availability).

### 4.7. Analysis of Differentially Expressed Genes

After filtering, the cells were grouped into clusters using the Leiden algorithm from the Seurat package. DEGs were identified using the FindMarkers function in Seurat with the MAST algorithm [51]. Statistical thresholds for DEGs were set to a Log fold Change > 1 and an adjusted *p*-value < 0.05, corrected for multiple testing using the Benjamini–Hochberg method, and a minimal percent parameter of 0.5. DEGs were used for cluster annotation and identification of function-specific subgroups within clusters. Kyoto Encyclopedia of Genes and Genomes and Gene Ontology (GO) pathway enrichment analysis was performed to find the signaling pathways, biological processes, molecular functions, and active cellular components for specific cell clusters.

### 4.8. Copy Number Karyotyping of Aneuploid Cells

The SCEVAN package [52] was used to detect copy number alterations. T cells were used as a reference. Cells that were labeled by the algorithm as aneuploid were considered to be tumor cells.

To test for possible false positives/negatives in CNV calling, repeats of the CNA analysis using the SCEVAN package with the provided ‘normal’ reference (represented by selected annotated T cells) and in the absence of the provided ‘normal’ reference (automatic selection of normal-like cells) were conducted. The convergence rate between data in these annotations was 97.16289% using the Jaccard index, which constitutes a high level of convergence for results of tumor population detection regardless of reference selection. Considering this data, the likelihood of false positives/negatives in CNV calling can be considered minimal. Additional cluster-level confidence metrics would not be informative due to SCEVAN integrating CNV signals across the entire genome instead of relying on localized/cluster-level statistical thresholds.

### 4.9. Cell–Cell Interaction Analysis

Ligand–receptor interactions between cells were analyzed using the CellChat algorithm [53]. Ligand–receptor interactions with *p*-value < 0.05, Log2 Fold Change > 0, and cell pct >10% were considered significant. The evaluation of statistical significance was based on the default permutation test (random permutation of the cell labels 100 times, followed by re-calculation of interaction probability). The Jaccard index heatmap and the top pathway comparison, detailing how the minimum cell percentage cutoff and the Log Fold Change cutoff influence the top interactions, can be found in Appendix A (Sheet 4, CellChat DEG threshold analysis). Full data on all cell–cell interactions (probability, *p*-values) can be found in Appendix A (Sheet 1, prob_pval_LR). *p*-values follow CellChat’s built-in permutation framework (label shuffling), and no additional FDR layer was applied beyond CellChat’s defaults.

## Figures and Tables

**Figure 1 ijms-26-11420-f001:**
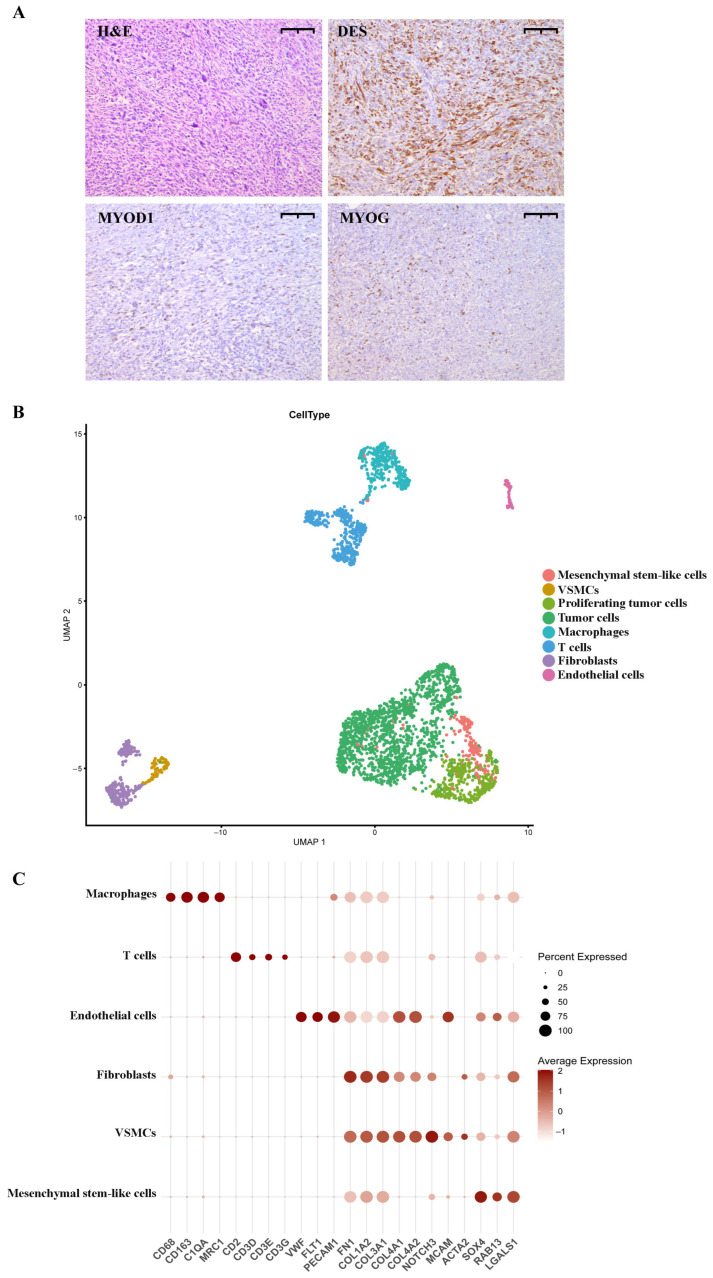
Histological and single-cell characterization of pRMS. (**A**) Immunohistochemistry of the pRMS (100× magnification, scale bar = 10 µm for all images). H&E: H&E staining of pRMS tumor tissue; DES: staining with anti-desmin antibodies; MYOD1: staining with anti-MYOD1 antibodies; MYOG: staining with anti-myogenin antibodies. (**B**) UMAP visualization of annotated pRMS clusters: mesenchymal stem-like cells—178 cells; VSMCs—78 cells; proliferating tumor cells—380 cells; tumor cells—1475 cells; macrophages—326 cells; T cells—344; fibroblasts—249 cells; and endothelial cells—69 cells. MAST algorithm; Log2 Fold Change > 1; adjusted *p*-value < 0.05; Benjamini–Hochberg method. The minimum percentage of cells in a cluster that express the gene is 0.5. (**C**) Dot plot showing marker gene expressions across annotated TME cells: macrophages—326 cells; T cells—344 cells; endothelial cells—69 cells; fibroblasts—249 cells; VSMCs—78 cells; and mesenchymal stem-like cells—178 cells. MAST algorithm; Log2 fold change > 1; adjusted *p*-value < 0.05; Benjamini–Hochberg method. The minimum percentage of cells in a cluster that express the gene is 0.5.

**Figure 2 ijms-26-11420-f002:**
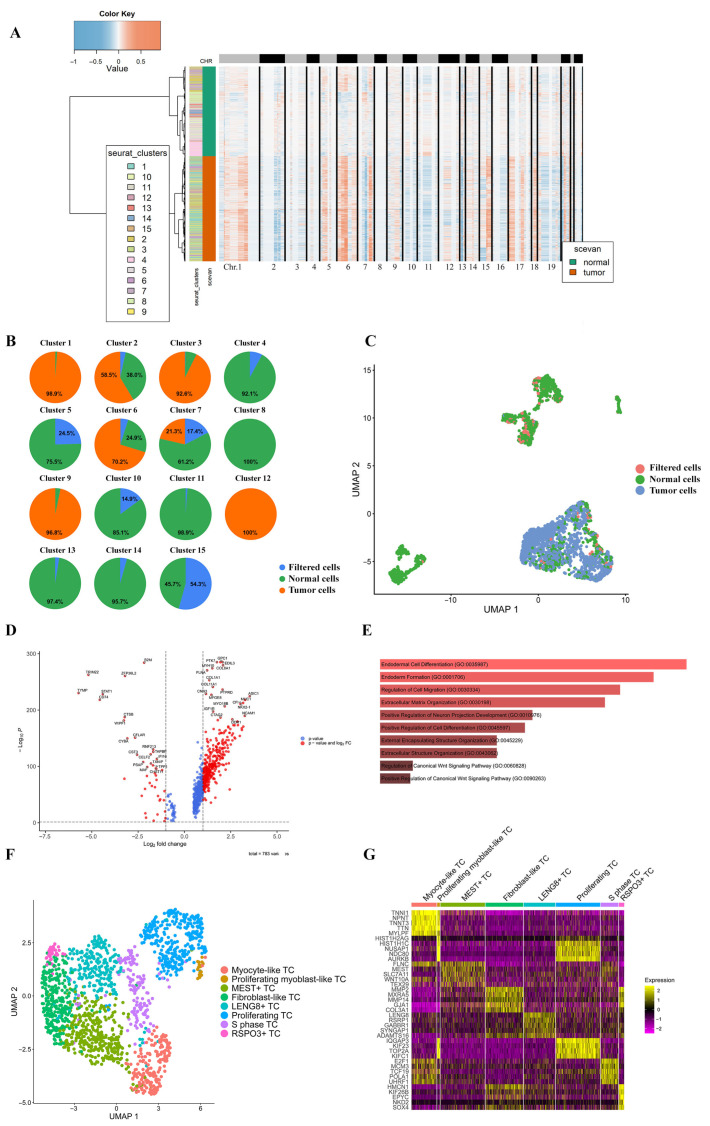
Characterization of pRMS tumor cells. (**A**) pRMS CNV heatmap by cluster. Chr., chromosome. (**B**) Pie charts of filtered, normal, and aneuploid (tumor) cells per cell cluster. (**C**) UMAP visualization of cell ploidy in pRMS cell populations. (**D**) Volcano plot depicting upregulation and downregulation of genes in tumor cells compared to the TME. MAST algorithm; Log2 Fold Change > 1; adjusted *p*-value < 0.05; Benjamini–Hochberg method. The minimum percentage of cells in a cluster that express the gene is 0.5. (**E**) Top 10 GO processes enriched in the majority of tumor cells (>70%, 1277/1825 cells) compared to the TME (adj. *p*-value < 0.05; sorted by *p*-value ranking). MAST algorithm; Log2 Fold Change > 1; adjusted *p*-value < 0.05; Benjamini–Hochberg method. The minimum percentage of cells in a cluster that express the gene is 0.7. (**F**) UMAP visualization of pRMS TCs: myocyte-like TC—191 cells; proliferating myoblast-like TC—21 cells; MEST^+^ TC—339 cells; fibroblast-like TC—287 cells; LENG8^+^ TC—241 cells; proliferating TC—337 cells; S-phase TC—135 cells; and RSPO3^+^ TC—40 cells. (**G**) A heatmap depicting TOP 5 differentially expressed genes in pRMS TCs (adj. *p*-value < 0.05; sorted by *p*-value ranking). MAST algorithm; Log2 Fold Change > 1; adjusted *p*-value < 0.05; Benjamini–Hochberg method. The minimum percentage of cells in a cluster that express the gene is 0.5.

**Figure 3 ijms-26-11420-f003:**
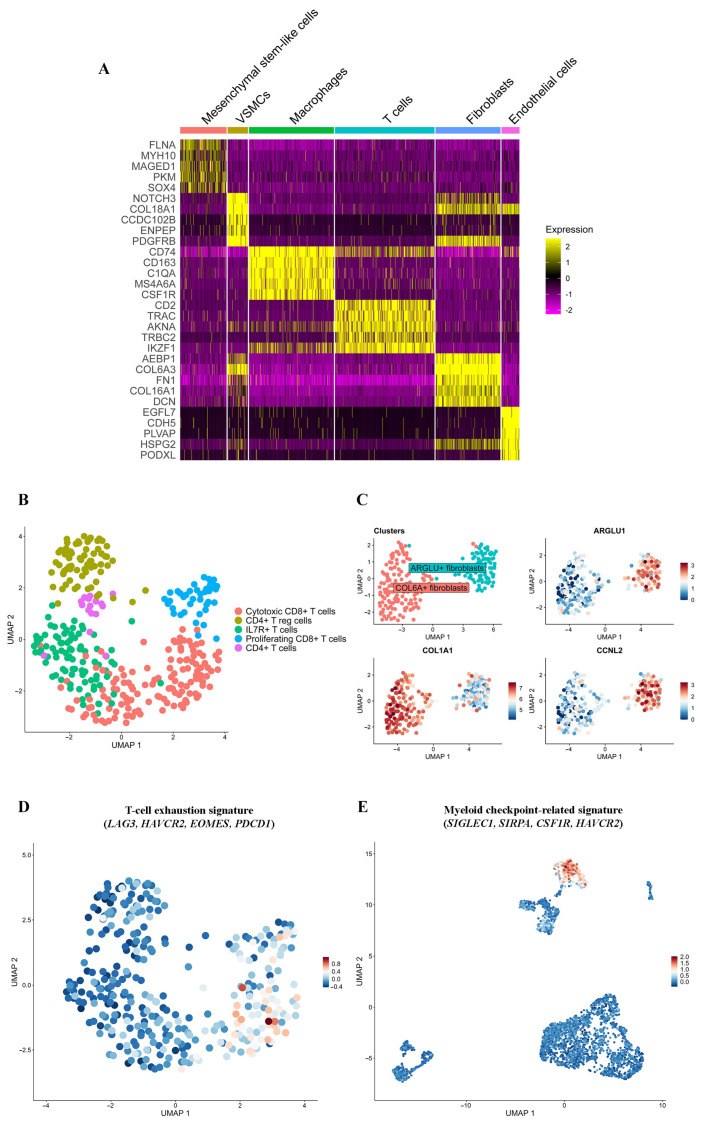
Characterization of the pRMS microenvironment. (**A**) Heatmap of TOP 5 differentially expressed genes in TME cell types. MAST algorithm; Log2 Fold Change > 1; adjusted *p*-value < 0.05; Benjamini–Hochberg method. The minimum percentage of cells in a cluster that express the gene is 0.5. (**B**) Annotated T-cell clusters: cytotoxic CD8^+^ T cells—131 cells; CD4^+^ T reg cells—69 cells; IL7R^+^ T cells—88 cells; proliferating CD8^+^ T cells—40 cells; and CD4^+^ T cells—16 cells. (**C**) Annotated fibroblast clusters. (**D**) UMAP plots showing the exhaustion signature (*LAG3*, *HAVCR2*, *EOMES*, *PDCD1*) in T-cell clusters. (**E**) UMAP plots showing the myeloid checkpoint-related signature (*SIGLEC1*, *SIRPA*, *CSF1R*, *HAVCR2*) in pRMS clusters.

**Figure 4 ijms-26-11420-f004:**
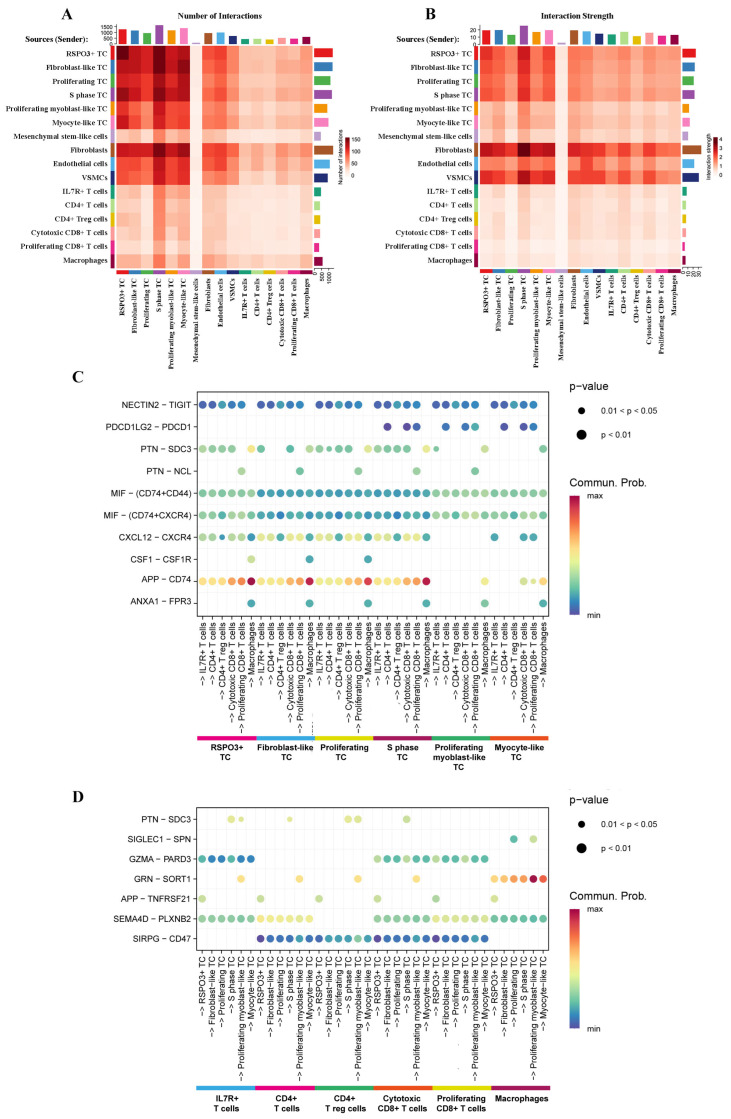
Cell–cell communication network in pRMS. (**A**) A heatmap showing the overall number of interactions between cell populations. (**B**) A heatmap showing the overall strength of interactions between cell populations. (**C**) Dot plot diagram depicting the signals with the highest communication probability sent from the TCs to the immune clusters. (**D**) Dot plot diagram depicting the signals with the highest communication probability sent from the immune clusters to the TCs. MAST algorithm; Log2 Fold Change > 0; adjusted *p*-value < 0.05; Benjamini–Hochberg method. The minimum percentage of cells in a cluster that express the gene is 0.1.

## Data Availability

The original data presented in the study are openly available in Zenodo under the accession number 17349024 (http://doi.org/10.5281/zenodo.17349024).

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
