# Peer review of "Single-Cell Sequencing Reveals Novel Tumor Populations and Their Interplay with the Immune Microenvironment in a Pleomorphic Rhabdomyosarcoma"

_ijms, 2025, doi:10.3390/ijms262311420_

Round 1
Reviewer 1 Report
Comments and Suggestions for Authors
This case study used single-cell transcriptomics to investigate the heterogeneous landscape of pRMS and tumor microenvironment. Some points should be noted as below,
- Given that although this disease is rare, there are also some multi-center case reports, it can be supplemented with some detailed information on prognosis and treatment.
- “These results suggest the potential of anti-MIF and anti-APP therapy as a strategy to help the immune cells in pRMS to recognize and attack cancer cells”, more direct experimental evidence should be supplemented; otherwise, it will be very difficult to explain this point.
- As the authors stated, this is the first time that single-cell sequencing has been used to conduct relevant research and analysis on this disease. However, we must clearly recognize that whether it is an attempt to analyze its microenvironment or the interactions between cells, its essence is based on the detection of gene and molecular expression. To put it more straightforwardly, no matter what kind of sequencing it is, its nature is a reductionist language, and its essence reflects the currently dominant "somatic mutation" theory. This can also be seen from the numerous genes listed in the research results. In fact, the old paradigm dominated by somatic and genetic mutations is hard to explain and solve complex clinical problems such as recurrence and drug resistance. It has not substantially improved the prognosis of patients, especially in the advanced stage. The following viewpoints (https://pubmed.ncbi.nlm.nih.gov/40443342/) can be referred to for appropriateIf we re-examine cancer from the perspectives of ecology and evolutionary biology, we may gain brand-new insights and solutions.
- Additionally, the authors may find it beneficial to consult the oncology monograph *Rethinking Cancer*. I believe this will be of great help to our comprehensive understanding of cancer and to future research and practice.
Author Response
Comments and Suggestions for Authors
This case study used single-cell transcriptomics to investigate the heterogeneous landscape of pRMS and tumor microenvironment. Some points should be noted as below,
Comment 1: Given that although this disease is rare, there are also some multi-center case reports, it can be supplemented with some detailed information on prognosis and treatment.
Response 1: Thank you for this valuable critique. Two multicenter studies of pRMS by Furlong et al. (2001) and Noujaim et al (2015) are already cited in the Introduction of our manuscript (Lines 33, 40-41). We have added a new citation of the population-based cohort study that discusses the decreased chemosensitivity of pRMS (Kobayashi et al, 2024) (Line 38-39). We have added a citation of the study focused on expression of PD-1/PD-L1 in pRMS (Torabi et al, 2017) (Line 43) and a case study that reveals the effectiveness of combining immunotherapy and standard chemotherapy to treat pRMS (Kournoutas et al, 2024) (Line 43-44). Information from the multicenter study (Poumeaud et al, 2024) on the link between the Lynch Syndrome and pRMS has also been added (Lines 52-53).
Comment 2: “These results suggest the potential of anti-MIF and anti-APP therapy as a strategy to help the immune cells in pRMS to recognize and attack cancer cells”, more direct experimental evidence should be supplemented; otherwise, it will be very difficult to explain this point.
Response 2: We agree that this statement in the abstract contains several points that need further evidence.The results we have described in this case report is all the data we have at the moment, so, to temper our claims, we have changed the aforementioned sentence to the following statement: “These results suggest that targeting MIF and APP signaling in pRMS may have therapeutic potential, although further investigation is needed.” (Lines 25-26).
Comment 3: As the authors stated, this is the first time that single-cell sequencing has been used to conduct relevant research and analysis on this disease. However, we must clearly recognize that whether it is an attempt to analyze its microenvironment or the interactions between cells, its essence is based on the detection of gene and molecular expression. To put it more straightforwardly, no matter what kind of sequencing it is, its nature is a reductionist language, and its essence reflects the currently dominant "somatic mutation" theory. This can also be seen from the numerous genes listed in the research results. In fact, the old paradigm dominated by somatic and genetic mutations is hard to explain and solve complex clinical problems such as recurrence and drug resistance. It has not substantially improved the prognosis of patients, especially in the advanced stage. The following viewpoints ) can be referred to for appropriateIf we re-examine cancer from the perspectives of ecology and evolutionary biology, we may gain brand-new insights and solutions.
Response 3: Thank you for this insightful commentary and the literature recommendation. We understand that our study has a number of limitations, being based on the results of single cell transcriptomic sequencing from one pRMS patient. We have added a paragraph in the Discussion section of our paper (Lines 374-382), in which we talk about the limitations of our case report. In this paragraph we also emphasize the importance of deciphering the tumor ecology and evolutionary biology of pRMS.
Comment 4: Additionally, the authors may find it beneficial to consult the oncology monograph *Rethinking Cancer*. I believe this will be of great help to our comprehensive understanding of cancer and to future research and practice.
Response 4: Thank you for the recommendation. We have consulted the oncology monograph *Rethinking Cancer*, added the commentary about tumor ecology to the limitations paragraph with the citation to the original source (Lines 379-382).
Reviewer 2 Report
Comments and Suggestions for Authors
1) Brief Summary
This case report performs single‑cell RNA‑seq (10x Genomics Fixed RNA Profiling) on a resected pleomorphic rhabdomyosarcoma (pRMS) from a 62‑year‑old female. The dataset includes 3,599 cells (median ~10,814 reads/cell; ~2,934 genes/cell; ~5,499 UMIs/cell; ~91.9% mapped reads). Leiden/UMAP clustering yields 15 clusters spanning multiple tumor and tumor‑microenvironment (TME) populations. Aneuploidy inference (SCEVAN) is used to demarcate malignant cells. Eight tumor subclusters are described (e.g., RSPO3+, fibroblast‑like, proliferating, S‑phase, proliferating myoblast‑like, myocyte‑like, MEST+, LENG8+). T cells partition into CD4+, Treg, proliferating CD8+, cytotoxic CD8+, and IL7R+ states; macrophages display immunosuppressive/TAM signatures. CellChat‑based analyses highlight putative tumor to immune signaling (notably MIF–CD74; in non‑myogenic clusters, APP–CD74, PTN–SDC3, CXCL12–CXCR4). The authors discuss anti‑MIF and anti‑APP as potential therapeutic hypotheses.
2) Overall Assessment
The topic is relevant, given the rarity of pRMS and the scarcity of single‑cell data. The manuscript is generally well structured and integrates histology/IHC, scRNA‑seq, CNV inference, and ligand–receptor predictions. However, several issues limit rigor and translational credibility: (i) all results derive from a single case without orthogonal validation; (ii) CellChat predictions are at times over‑interpreted as therapeutic recommendations; (iii) methodological/reporting inconsistencies; and (iv) a tone around ‘targetability’ stronger than the data support. Overall recommendation: Major Revision.
3) Strengths
- Rare entity with clinically annotated timeline and histology/IHC confirmation (DES+, focal MYOG/MYOD1).
- Reasonable fixed‑cell scRNA‑seq quality; explicit QC thresholds; doublet detection; CNV‑based tumor identification.
- Coherent annotation of diverse tumor states (myogenic vs non‑myogenic) and immune/stromal populations with plausible signatures.
- Differential‑expression thresholds and enrichment analyses are stated; Methods and marker tables are detailed and helpful.
4) Major Comments
4.1 Single‑patient scope and generalisability
All findings come from a single specimen. Please temper claims (especially therapeutic implications) and clearly separate observation from hypothesis. Where feasible, corroborate key findings with orthogonal assays (IHC/IF or spatial transcriptomics) on the same block or an independent pRMS case.
4.2 Over‑interpretation of CellChat predictions
Ligand–receptor inferences (MIF–CD74; APP–CD74; PTN–SDC3; CXCL12–CXCR4) are transcript‑level predictions. No protein‑level validation or perturbation is shown. Present these as hypotheses, add IHC/IF for relevant proteins with co‑localisation where possible, or substantially soften therapeutic language.
4.3 Questionable inclusion of CD74 as a T‑cell exhaustion marker
The text lists CD74 among exhaustion markers in cytotoxic CD8+ T cells; CD74 is the MHC‑II invariant chain, typically on B cells/myeloid cells, not a canonical T‑cell exhaustion gene. Please revisit the marker list and adjust narrative/plots accordingly.
4.4 Macrophage checkpoint nomenclature
‘CSFR1’ appears where ‘CSF1R’ seems intended; elsewhere CSF1R is used. Standardise nomenclature across text/figures.
4.5 Definition/validation of tumor vs TME compartments
Tumor classification relies on inferred CNV (SCEVAN) with T cells as reference. Add representative CNV heatmaps by cluster, report malignant fraction per cluster, overlay copy‑number scores on UMAPs, and provide per‑cluster aneuploidy percentages; discuss potential false positives/negatives (stromal aneuploidy, ambient RNA).
4.6 Reproducibility of DEG thresholds and internal consistency
Results sometimes filter DEGs with pct > 0.7 while Methods state min.pct = 0.5. Ensure consistent logFC/min.pct/adjusted‑p thresholds across sections/figures and specify Seurat/MAST settings per comparison.
4.7 QC choices for fixed‑cell data
The mitochondrial‑read filter (mt% < 5%) may be strict for fixed‑cell protocols. Justify with distributions per cluster. Describe ambient RNA handling (e.g., SoupX) and provide a QC summary table (median nFeature/nCount by cluster; doublet rates; % filtered).
4.8 Therapeutic inferences require caution
Proposals like anti‑MIF/anti‑APP are interesting but speculative in a single case. Given PD‑(L)1 signaling was not detected by CellChat, discuss possible technical false negatives and avoid strong negative mechanistic claims. Recast as ‘testable hypotheses.’
4.9 Figure/legend completeness
For UMAP and dot/volcano plots, include legends with N (cells/clusters), statistical tests, multiple‑testing correction, and thresholds. For histology/IHC, add scale bars/magnifications; standardise gene symbols and pathway names.
4.10 Data availability and minimal reproducible package
If raw data cannot be public, provide de‑identified processed matrices (gene×cell), cluster annotations, and analysis scripts/notebooks under controlled access or as Supplementary files to enable reproducibility.
5) Minor Comments
- Standardise subtype/cluster naming (e.g., ‘myocyte‑like’, ‘RSPO3+’) across figures and text; avoid mixing exhaustion and activation markers.
- Report exact cell counts for each cluster (% of total) and add a compact assay summary table (platform, read length, reference genome).
- Clarify whether the 10x Fixed RNA protocol was used end‑to‑end and the fixation timing relative to surgery; note any cold‑ischemia time.
- For T‑cell states, distinguish stem‑like (TCF7/IL7R) vs cytotoxic (GZMB/GZMH) vs exhausted (LAG3/HAVCR2/PDCD1) using standard references.
- Spell‑check gene names (e.g., ‘CSF1R’, not ‘CSFR1’; ‘TNNT2’ etc.) and keep gene symbols consistently formatted.
- In macrophages, specify marker panels/thresholds for TAM subsets and provide their proportions.
- State how doublets (including potential tumor–stromal hybrids) were identified and removed.
- Note any batch effects (library lanes) and the integration/normalisation strategy (e.g., SCTransform parameters).
- Add explicit citations where claims are borrowed (e.g., GZMK‑high CD8 subsets), and map them to your data in plots.
- Light English polishing: unify decimal notation, p‑value formatting, and minor typographical issues.
6) Recommendation
Major Revision — The study offers a valuable single‑cell snapshot of an exceedingly rare sarcoma subtype, but key methodological and interpretive issues must be addressed. With tempered conclusions, clearer QC/DEG reporting, orthogonal validation of key axes (MIF–CD74, APP–CD74, PTN–SDC3, CXCL12–CXCR4), and improved figure/legend completeness, the manuscript could progress toward publication.
Author Response
Comments and Suggestions for Authors
1) Brief Summary
This case report performs single‑cell RNA‑seq (10x Genomics Fixed RNA Profiling) on a resected pleomorphic rhabdomyosarcoma (pRMS) from a 62‑year‑old female. The dataset includes 3,599 cells (median ~10,814 reads/cell; ~2,934 genes/cell; ~5,499 UMIs/cell; ~91.9% mapped reads). Leiden/UMAP clustering yields 15 clusters spanning multiple tumor and tumor‑microenvironment (TME) populations. Aneuploidy inference (SCEVAN) is used to demarcate malignant cells. Eight tumor subclusters are described (e.g., RSPO3+, fibroblast‑like, proliferating, S‑phase, proliferating myoblast‑like, myocyte‑like, MEST+, LENG8+). T cells partition into CD4+, Treg, proliferating CD8+, cytotoxic CD8+, and IL7R+ states; macrophages display immunosuppressive/TAM signatures. CellChat‑based analyses highlight putative tumor to immune signaling (notably MIF–CD74; in non‑myogenic clusters, APP–CD74, PTN–SDC3, CXCL12–CXCR4). The authors discuss anti‑MIF and anti‑APP as potential therapeutic hypotheses.
2) Overall Assessment
The topic is relevant, given the rarity of pRMS and the scarcity of single‑cell data. The manuscript is generally well structured and integrates histology/IHC, scRNA‑seq, CNV inference, and ligand–receptor predictions. However, several issues limit rigor and translational credibility: (i) all results derive from a single case without orthogonal validation; (ii) CellChat predictions are at times over‑interpreted as therapeutic recommendations; (iii) methodological/reporting inconsistencies; and (iv) a tone around ‘targetability’ stronger than the data support. Overall recommendation: Major Revision.
3) Strengths
- Rare entity with clinically annotated timeline and histology/IHC confirmation (DES+, focal MYOG/MYOD1).
- Reasonable fixed‑cell scRNA‑seq quality; explicit QC thresholds; doublet detection; CNV‑based tumor identification.
- Coherent annotation of diverse tumor states (myogenic vs non‑myogenic) and immune/stromal populations with plausible signatures.
- Differential‑expression thresholds and enrichment analyses are stated; Methods and marker tables are detailed and helpful.
4) Major Comments
Comment 1: Single‑patient scope and generalisability
All findings come from a single specimen. Please temper claims (especially therapeutic implications) and clearly separate observation from hypothesis. Where feasible, corroborate key findings with orthogonal assays (IHC/IF or spatial transcriptomics) on the same block or an independent pRMS case.
Response 1: Thank you for this critique. To temper our claims, especially when talking about potential therapy, we have changed statements in the Abstract (Lines 25-26), Results (Lines 168, 172-173, 267-268, 273-275, 278-280, 292-293, 298, 307, 315-316) and Discussion (Lines 339-342, 345-347, 353-355, 361-369) We have also made a separate paragraph in the Discussion section (Lines 374-382), which outlines the limitations of our case study.
Unfortunately, the tumor sample for this case report was very rare. At this moment we don’t have another pRMS case or the opportunity to perform IHC/IF or spatial transcriptomics on tissue from the same sample.
Comment 2: Over‑interpretation of CellChat predictions
Ligand–receptor inferences (MIF–CD74; APP–CD74; PTN–SDC3; CXCL12–CXCR4) are transcript‑level predictions. No protein‑level validation or perturbation is shown. Present these as hypotheses, add IHC/IF for relevant proteins with co‑localisation where possible, or substantially soften therapeutic language.
Response 2: Yes, we agree with this critique. We have softened the therapeutic language and presented our cell-cell interaction findings as hypothetical, where possible. The following lines have been altered: Lines 292-293, 298, 307, 315-316, 339-342, 345-347, 361-369, 386-387.
Unfortunately, the tumor sample for this case report was very rare. At this moment we don’t have another pRMS case or the opportunity to perform IHC/IF or spatial transcriptomics on tissue from the same sample.
Comment 3: Questionable inclusion of CD74 as a T‑cell exhaustion marker
The text lists CD74 among exhaustion markers in cytotoxic CD8+ T cells; CD74 is the MHC‑II invariant chain, typically on B cells/myeloid cells, not a canonical T‑cell exhaustion gene. Please revisit the marker list and adjust narrative/plots accordingly.
Response 3: Yes, we agree with this correction. We have removed CD74 from the exhaustion marker lists from the text and figure legends, and reuploaded the UMAP plots depicting the expression of the modified exhaustion signature in T cells and macrophages (Fig. 3D, Fig. 3E).
Comment 4: Macrophage checkpoint nomenclature
‘CSFR1’ appears where ‘CSF1R’ seems intended; elsewhere CSF1R is used. Standardise nomenclature across text/figures.
Response 4: Thank you for notifying us of this discrepancy. We have corrected the mentions of ‘CSFR1’ to ‘CSF1R’ in the text and figures.
Comment 5: Definition/validation of tumor vs TME compartments
Tumor classification relies on inferred CNV (SCEVAN) with T cells as reference. Add representative CNV heatmaps by cluster, report malignant fraction per cluster, overlay copy‑number scores on UMAPs, and provide per‑cluster aneuploidy percentages; discuss potential false positives/negatives (stromal aneuploidy, ambient RNA).
Response 5: Thank you for this valuable critique. We have added a CNA matrix heatmap from SCEVAN analysis to the Supplementary Files (Table S3. QC metrics, identification of tumor cells, cell counts). We have also added the report of malignant fraction per each cell cluster to the Supplementary Files (Table S3. QC metrics, identification of tumor cells, cell counts). The malignant fraction of cells per cluster we identified as tumor clusters is also reported in the text (Lines 134-135). To the best of our knowledge, the overlay of copy-number scores on UMAPs is not technically feasible with the SCEVAN toolkit.
The levels of ambient RNA in our dataset were not elevated, so in our opinion it did not contribute to false positives/negatives. The stromal aneuploidy was minimal, with a few cells annotated as stromal having aneuploidy. These cells were filtered out. Again, we do not consider their contribution to false positives/negatives to be substantial.
Comment 6: Reproducibility of DEG thresholds and internal consistency
Results sometimes filter DEGs with pct > 0.7 while Methods state min.pct = 0.5. Ensure consistent logFC/min.pct/adjusted‑p thresholds across sections/figures and specify Seurat/MAST settings per comparison.
Response 6: Thank you for pointing this out. The DEG analysis settings in this paper are: Log2 Fold Change >1, adjusted p-value <0.05, minimal percent parameter of 0.5. We've corrected the Lines 137-138 to read “Gene expression levels were compared between six tumor and TME cell clusters (Log2 Fold Change >1, pct. >0.5, adj. p-value <0.05; Figure 2B).” However, in this section we also wanted to point out the upregulated genes for the majority of tumor cells (more than 70%). We modified the following sentence (Lines 138-139) to make this point more clear.
Comment 7: QC choices for fixed‑cell data
The mitochondrial‑read filter (mt% < 5%) may be strict for fixed‑cell protocols. Justify with distributions per cluster. Describe ambient RNA handling (e.g., SoupX) and provide a QC summary table (median nFeature/nCount by cluster; doublet rates; % filtered).
Response 7: Thank you for this valuable critique. We added a diagram of the distribution of Seurat quality control metrics (nFeature RNA, nCount RNA, percent.mt) by cluster to justify the use of the mt% < 5% threshold to the Supplementary Files (Table S3. QC metrics, identification of tumor cells, cell counts.).Thank you for the suggestion to use the SoupX package. We considered its use, but based on the results of the differential expression analysis, we decided not to use SoupX to remove RNA contamination, since the data did not contain elevated values of contaminating RNAs and we wanted to try to avoid possible effects of expression matrix correction on data interpretation. We have added a quality control summary table (median nFeature/nCount per cluster) to the Supplementary Files (Table S3. QC metrics, identification of tumor cells, cell counts.) and information about filtered cells and duplicate rates to the Line 122 and Line 440.
Comment 8: Therapeutic inferences require caution
Proposals like anti‑MIF/anti‑APP are interesting but speculative in a single case. Given PD‑(L)1 signaling was not detected by CellChat, discuss possible technical false negatives and avoid strong negative mechanistic claims. Recast as ‘testable hypotheses.’
Response 8: Thank you for this critique. According to previous case studies, PDL1 is not expressed in all cases of pRMS. We added Line 43 to the Introduction to include the paper, in which this phenomenon is mentioned. In the Discussion section we added Lines 360-361 to mention this tendency and how it applies to our results. So we do not have grounds to consider the absence of PD‑(L)1 signaling detection by CellChat a false negative. However, we have softened the therapeutic language when discussing CellChat results, as we have mentioned in the answer to the comment in the Section 4.2.
Comment 9: Figure/legend completeness
For UMAP and dot/volcano plots, include legends with N (cells/clusters), statistical tests, multiple‑testing correction, and thresholds. For histology/IHC, add scale bars/magnifications; standardise gene symbols and pathway names.
Response 9: Thank you for this valuable commentary. For UMAP and Dot Plots, we have added the number of cells per cluster to the legends (Lines 109-111, 113-115, 154-156, 232-233). We also have added the statistical test names, thresholds, and multiple-testing correction to the figure legends (Lines 111-113, 115-117, 149-150, 152-154, 157-159, 230-232, 323-324). For histology we have added a scale bar to the images (Figure 1A) and scale bar/magnifications to the figure legend (Line 106). Gene symbols and pathway names have been standardized.
Comment 10: Data availability and minimal reproducible package
If raw data cannot be public, provide de‑identified processed matrices (gene×cell), cluster annotations, and analysis scripts/notebooks under controlled access or as Supplementary files to enable reproducibility.
Response 10: Thank you for mentioning this point. We have added information Lines 475-476 that the data and scripts used in the analysis will be available upon request through e-mail to the corresponding author.
5) Minor Comments
Comment 11: Standardise subtype/cluster naming (e.g., ‘myocyte‑like’, ‘RSPO3+’) across figures and text; avoid mixing exhaustion and activation markers.
Response 11: Thank you for this critique. We have standardized the subtype and cluster names and checked the accuracy of exhaustion and activation markers throughout our text and figures.
Comment 12: Report exact cell counts for each cluster (% of total) and add a compact assay summary table (platform, read length, reference genome).
Response 12: Thank you for pointing this out. We have added the cell counts for each cluster to the Supplementary files (Table S3. QC metrics, identification of tumor cells, cell counts.). The information on the sequencing platform, read length, and reference genome is available in the Methods section (Lines 428-430, 433).
Comment 13: Clarify whether the 10x Fixed RNA protocol was used end‑to‑end and the fixation timing relative to surgery; note any cold‑ischemia time.
Response 13: Thank you for pointing this out. The fixation of the tumor sample was conducted immediately after the excision during surgery. We have added a clarifying statement about this in Lines 414-415.
Comment 14: For T‑cell states, distinguish stem‑like (TCF7/IL7R) vs cytotoxic (GZMB/GZMH) vs exhausted (LAG3/HAVCR2/PDCD1) using standard references.
Response 14: Thank you for this critique. We have consulted the article on the interpretation of T cell states using reference reference atlases (Andreatta et al, 2021) to answer this question. We have added Lines 225-227 to clarify that exhausted state T cell state coincided with the cytotoxic T cell state for pRMS T cells, while the stem-like T cells did not demonstrate any exhaustion. We have also added the T cell state signatures to the Supplementary files (Table S1. Markers and gene signatures.).
Comment 15: Spell‑check gene names (e.g., ‘CSF1R’, not ‘CSFR1’; ‘TNNT2’ etc.) and keep gene symbols consistently formatted.
Response 15: Thank you for pointing this out. We have spellchecked gene names for consistency throughout the manuscript.
Comment 16: In macrophages, specify marker panels/thresholds for TAM subsets and provide their proportions.
Response 16: Thank you for this critique. We have added the marker signature for the anti-inflammatory cysteine cathepsin-positive TAMs (Wei et al, 2023) that we used to characterize the pRMS macrophages to the Supplementary files (Table S1. Markers and gene signatures.). We have also modified the Lines 238-240 to make it clear that the pRMS macrophages express key genes from this signature, but do not necessarily equal the APOE+CTSZ+ TAMs described by Wei et al.
Comment 17: State how doublets (including potential tumor–stromal hybrids) were identified and removed.
Response 17: Thank you for this critique. We added additional information about the approach used to filter duplicates: “Doublet detection was performed using the DoubletCollection package [12]. Four additional packages were also used to identify doublets: 'DoubletFinder' [42], 'cxds' [43], 'scDblFinder [44], and 'scrublet [45]. Cells that were identified as doublets by at least one of the methods were considered doublets. The duplicate rate was 2.7%” (Lines 438-440).
Comment 18: Note any batch effects (library lanes) and the integration/normalisation strategy (e.g., SCTransform parameters).
Response 18: Thank you for this critique. Batch effects were absent due to the presence of a single sample in the sample set. SCTransform with vst.flavor="v2" was used for normalization. We have added this information to Line 440.
Comment 19: Add explicit citations where claims are borrowed (e.g., GZMK‑high CD8 subsets), and map them to your data in plots.
Response 19: Thank you for this critique. We have checked the citations, where we discuss data acquired previously. New citations have been added to Lines 173, 251, 259.
Comment 20: Light English polishing: unify decimal notation, p‑value formatting, and minor typographical issues.
Response 20: Thank you for this critique. We have checked the text for unified decimal notation, p‑value formatting, and minor typographical issues.
6) Recommendation
Major Revision — The study offers a valuable single‑cell snapshot of an exceedingly rare sarcoma subtype, but key methodological and interpretive issues must be addressed. With tempered conclusions, clearer QC/DEG reporting, orthogonal validation of key axes (MIF–CD74, APP–CD74, PTN–SDC3, CXCL12–CXCR4), and improved figure/legend completeness, the manuscript could progress toward publication.
Response: Thank you for such detailed and helpful critique. We hope that the added changes to the overall language of the text, improved images, additional QC and cell count data, and additional references made our study better and easier to understand.
Reviewer 3 Report
Comments and Suggestions for Authors
This study is the first to demonstrate an inhibitory interaction between tumor and immune cells in scRNA-seq analysis of pRMS, and it is intriguing that reversing this interaction may improve anti-tumor efficacy. However, the following points should be added: (1) Tumor immune response. This case suggests that inhibitors of MIF and APP are potential targets for the treatment of pRMS. In this case, please also mention in the discussion section that MIF is distributed primarily in all pRMS tumors, and APP is distributed primarily in non-myogenic pRMS tumors. (2) Pleomorphic rhabdomyosarcoma is a rare soft tissue tumor that lies between rhabdomyosarcoma and undifferentiated pleomorphic sarcoma. Please explain why this is intermediate. (3) Please explain why non-myogenic clusters have been found to have more communication links with immune populations. (4) Please summarize and add any additional insights gained from this study regarding the mechanisms of pRMS and the molecular characteristics of tumor and immune cell populations.
Author Response
Comments and Suggestions for Authors
This study is the first to demonstrate an inhibitory interaction between tumor and immune cells in scRNA-seq analysis of pRMS, and it is intriguing that reversing this interaction may improve anti-tumor efficacy.
However, the following points should be added:
Comment 1: Tumor immune response. This case suggests that inhibitors of MIF and APP are potential targets for the treatment of pRMS. In this case, please also mention in the discussion section that MIF is distributed primarily in all pRMS tumors, and APP is distributed primarily in non-myogenic pRMS tumors.
Response 1: Thank you for this commentary. Our case study was focused on cell populations from one pRMS tumor. According to our cell-cell communication analysis, MIF signalling is initiated by all the tumor clusters in this pRMS sample, while APP signalling has a strong probability of happening between non-myogenic clusters and the immune cells. We have modified Lines 362-364 in the Discussion section to make this point more clear.
Comment 2: Pleomorphic rhabdomyosarcoma is a rare soft tissue tumor that lies between rhabdomyosarcoma and undifferentiated pleomorphic sarcoma. Please explain why this is intermediate.
Response 2: Thank you for this valuable critique. We use this statement due to pRMS displaying both pleomorphic and skeletal muscle features, which is detailed in previous studies (Fan et al, 2022, Agaram et al, 2022). The references for these studies are provided in Line 33 of Introduction. Additionally, in our study we describe the specific myogenic and non-myogenic (pleomorphic) tumor clusters, which potentially play different roles in tumor progression. This gives us additional grounds to say that pRMS combines qualities of RMS and UPS. We have added Lines 326-328 to make this point more clear.
Comment 3: Please explain why non-myogenic clusters have been found to have more communication links with immune populations.
Response 3: Thank you for this interesting question. The fact that the non-myogenic clusters and pRMS have stronger and more numerous ties to the immune cells than the myogenic clusters is experimental data we have acquired through Cellchat analysis of cell-cell communications. While we can speculate about the specific pathways in the text, we do not yet understand the exact reason why this overall tendency exists. We think this phenomenon needs further study. We have added Line 341 to explain this in the Discussion section.
Comment 4: Please summarize and add any additional insights gained from this study regarding the mechanisms of pRMS and the molecular characteristics of tumor and immune cell populations.
Response 4: Thank you for this commentary. We summarize our findings on the pRMS biology that can be learned from this case study in the Discussion section of our paper. We discuss what we have learned about tumor populations, about the TME, and about the communications between tumor and immune cells. Additionally, we have added a paragraph about the limitations and future directions that can be explored to learn more about this rare tumor (Lines 374-382).
Reviewer 4 Report
Comments and Suggestions for Authors
This study is a descriptive case report, well-presented and interesting, and merits publication. However, although it may seem redundant, it is suggested that consent and approval be added as the first subheading in the methods section.
For example:
4.1. Consent and Approval
The written informed consent to clinical examination and the publication of their details was obtained. This study was performed in accordance with the Declaration of Helsinki and approved by the Ethics Committee of N.N. Blokhin National Medical Research Center of Oncology (Approval number: 2020-42).
Author Response
Comments and Suggestions for Authors
Comment 1: This study is a descriptive case report, well-presented and interesting, and merits publication. However, although it may seem redundant, it is suggested that consent and approval be added as the first subheading in the methods section.
For example:
4.1. Consent and Approval
The written informed consent to clinical examination and the publication of their details was obtained. This study was performed in accordance with the Declaration of Helsinki and approved by the Ethics Committee of N.N. Blokhin National Medical Research Center of Oncology (Approval number: 2020-42).
Response 1: Thank you for this suggestion. As suggested, we have added the consent and approval as the first subheading in the methods section (Lines 389-393).
Round 2
Reviewer 2 Report
Comments and Suggestions for Authors
1) Brief Summary
This single‑case report applies fixed‑cell single‑cell RNA‑sequencing to a pleomorphic rhabdomyosarcoma (pRMS), identifying multiple tumor and immune subsets, inferring copy‑number alterations to separate malignant from microenvironmental cells, and exploring putative ligand–receptor signalling (e.g., tumor myeloid/T‑cell axes). Version v2 revises language around therapeutic implications, adds QC summaries and supplementary CNV visualisations, and expands figure legends.
2) What Changed from v1 to v2 (per manuscript and response)
- Therapeutic statements reframed as hypothesis‑generating; limitations section strengthened.
- Quality‑control metrics consolidated; rationale for mitochondrial read cut‑off added; supplementary CNV heatmaps and malignant‑fraction summaries provided.
- Figure legends improved (N, tests, thresholds), and histology figures annotated with scale bars.
- Data/code availability clarified as ‘on request’ rather than public repository deposition.
3) Did v2 Resolve First‑Round Concerns? (Itemised)
3.1 Therapeutic over‑interpretation
Partially resolved. The tone is more cautious; however, several mechanistic inferences still read prescriptive without protein‑level validation. Please keep all treatment‑oriented text explicitly hypothesis‑generating.
3.2 Single‑patient scope and external validity
Unresolved. Conclusions still rest on one specimen; no orthogonal validation or second case is provided. This must be emphasised in the Abstract and Conclusions.
3.3 Marker panels (T‑cell exhaustion; macrophage checkpoints)
Partially resolved. Marker lists are cleaner, but ensure that non‑canonical markers (e.g., invariant‑chain genes) are not labelled as exhaustion without strong justification.
3.4 CNV‑guided tumor calling (SCEVAN) and reporting
Partially resolved. Supplementary heatmaps/malignant fractions are helpful; integrate a representative heatmap and a concise per‑cluster malignant fraction table into the main text for transparency.
3.5 QC/ambient RNA handling
Partially resolved. mt% thresholds are justified; ambient RNA/decontamination decisions would benefit from quantitative estimates or sensitivity checks.
3.6 Statistical reporting & robustness
Unresolved. No quantitative sensitivity analyses are shown for DEG thresholds or ligand–receptor inferences; effect sizes and adjusted p‑values need to be consistently reported.
3.7 Reproducibility and data access
Unresolved. ‘On‑request’ access is not equivalent to public deposition. Processed matrices, metadata, and scripts should be archived with a DOI if de‑identification permits.
4) Major Comments
- Protein‑level validation for key axes
At minimum, provide IHC/IF for selected ligand–receptor pairs central to the claims (e.g., tumor MIF with CD74‑positive myeloid/B‑cell compartments). If archival tissue is insufficient, explicitly state limitations and present the section as future work.
- Quantitative robustness of inferences
Report how varying DEG thresholds (e.g., min.pct, logFC) and CellChat parameters alters top interactions; include permutation/bootstrapping to estimate false‑positive rates.
- Primary vs Supplementary integration of CNV results
Move one CNV heatmap to the main figures and summarise malignant‑fraction per cluster in the Results; this is foundational to all downstream analyses.
- Clarify tumor vs TME classification errors
Discuss possible false positives/negatives in CNV calling (e.g., stromal aneuploidy, ambient RNA), and provide cluster‑level confidence metrics or QC flags.
- Reproducibility package
Deposit de‑identified gene×cell matrices, cluster annotations, and analysis scripts/notebooks in a public repository (Zenodo/Figshare/GEO) with an accession number cited in Data Availability.
- Tone and claims
Ensure all translational statements are hypothesis‑generating; avoid implying clinical actionability without orthogonal validation and replication.
5) Minor Comments
– Provide exact cell counts per cluster and their percentage of total; include a compact QC table (median nFeature/nCount per cluster; doublet rate; % filtered).
– Standardise gene symbol formatting (italics where required) and check for residual typos; unify British/American spelling across the manuscript.
– Specify the statistical test and multiple‑testing correction in every figure/table caption; report adjusted p‑values in Supplementary tables.
– Add/confirm scale bars on all histology images and ensure UMAP/dot‑plot font sizes are legible at print scale.
– State whether any batch effects (lanes/libraries) were present and how integration (e.g., SCTransform) handled them.
– Clarify doublet removal strategy and how potential tumor–stromal doublets were adjudicated.
– Define all abbreviations at first use (e.g., TAM, CAF, Treg); keep terminology consistent throughout.
6) Recommendation
Major Revision. Version2 is improved (clearer language, better QC/legend reporting, supplementary CNV visuals), but the study remains constrained by single‑case scope, absence of protein‑level validation, limited robustness analyses, and lack of public reproducibility materials. Addressing the above major points would bring the manuscript closer to a publishable standard.
Author Response
3) Did v2 Resolve First‑Round Concerns? (Itemised)
3.1 Therapeutic over‑interpretation
Partially resolved. The tone is more cautious; however, several mechanistic inferences still read prescriptive without protein‑level validation. Please keep all treatment‑oriented text explicitly hypothesis‑generating.
Answer: Thank you for this critique. We have further modified Lines 26-28, 79, 292-293, 296, 299, 300-301, 302, 305, 310, 315, 320-321, 372-375, 378-385, 391-392, 396-400, 403-405 to emphasize the transcriptional level of our data and the need for future protein-level validation, larger cohorts, and in vitro and in vivo validation in order to reach clinical actionability.
3.2 Single‑patient scope and external validity
Unresolved. Conclusions still rest on one specimen; no orthogonal validation or second case is provided. This must be emphasised in the Abstract and Conclusions.
Answer: Thank you for this critique. Lines 26-28 in the Abstract, Lines 383-385 in the Discussion and Lines 403-405 in the Conclusion have been modified to emphasize that we do not claim clinical actionability of our results. The added paragraph on future studies (Lines 396-400) emphasizes the need for larger cohorts, protein verification, and in vitro and in vivo validation in the future.
3.3 Marker panels (T‑cell exhaustion; macrophage checkpoints)
Partially resolved. Marker lists are cleaner, but ensure that non‑canonical markers (e.g., invariant‑chain genes) are not labelled as exhaustion without strong justification.
Answer: Thank you for this critique.
We have changed Line 152 to describe the mentioned genes (CD24, CDH2) as being involved with immune cell interactions, rather than being related to immune exhaustion specifically (Line152).
The genes that remain on our lists as markers of the exhausted T cell group are as follows: LAG3, HAVCR2, EOMES, PDCD1, TIGIT. Here we provide four recent articles that summarize the evidence of this role for these genes:
- Andreatta, M., et al., Interpretation of T cell states from single-cell transcriptomics data using reference atlases. Nat Commun, 2021. 12(1): p. 2965.(Reference 17 in our study)
- Nair R, Somasundaram V, Kuriakose A, Krishn SR, Raben D, Salazar R, Nair P. Deciphering T-cell exhaustion in the tumor microenvironment: paving the way for innovative solid tumor therapies. Front Immunol. 2025 Apr 1;16:1548234. doi: 10.3389/fimmu.2025.1548234.
- Zhang C, Sheng Q, Zhang X, Xu K, Jin X, Zhou W, Zhang M, Lv D, Yang C, Li Y, Xu J, Li X. Prioritizing exhausted T cell marker genes highlights immune subtypes in pan-cancer. iScience. 2023 Mar 24;26(4):106484. doi: 10.1016/j.isci.2023.106484.
- Li J, He Y, Hao J, Ni L, Dong C. High Levels of Eomes Promote Exhaustion of Anti-tumor CD8+ T Cells. Front Immunol. 2018 Dec 18;9:2981. doi: 10.3389/fimmu.2018.02981.
We have also changed the phrase “myeloid exhaustion” to “myeloid checkpoint-related” genes in Line 25, Line 245, and Figure 3E. The genes we list as pertaining to myeloid checkpoints are as follows: SIGLEC1, SIRPA, CSF1R, HAVCR2. Here we provide four recent review articles that support this role for these genes:
- Qian Y, Yang T, Liang H, Deng M. Myeloid checkpoints for cancer immunotherapy. Chin J Cancer Res. 2022 Oct 30;34(5):460-482. doi: 10.21147/j.issn.1000-9604.2022.05.07. .(Added as Reference 27 to our study – Line252)
- Zhao L, Cheng S, Fan L, Zhang B, Xu S. TIM-3: An update on immunotherapy. Int Immunopharmacol. 2021 Oct;99:107933. .(Added as Reference 28 in our study – Line252)
- Ma C, Li Y, Li M, Lv C, Tian Y. Targeting immune checkpoints on myeloid cells: current status and future directions. Cancer Immunol Immunother. 2025 Jan 3;74(2):40. doi: 10.1007/s00262-024-03856-6.
- Chan C, Lustig M, Baumann N, Valerius T, van Tetering G, Leusen JHW. Targeting Myeloid Checkpoint Molecules in Combination With Antibody Therapy: A Novel Anti-Cancer Strategy With IgA Antibodies? Front Immunol. 2022 Jul 5;13:932155. doi: 10.3389/fimmu.2022.932155.
3.4 CNV‑guided tumor calling (SCEVAN) and reporting
Partially resolved. Supplementary heatmaps/malignant fractions are helpful; integrate a representative heatmap and a concise per‑cluster malignant fraction table into the main text for transparency.
Answer: Thank you for this critique. We have integrated a representative CNV heatmap (Figure 2A) and a list of percluster malignant fractions (Lines 138-141) into the main text.
3.5 QC/ambient RNA handling
Partially resolved. mt% thresholds are justified; ambient RNA/decontamination decisions would benefit from quantitative estimates or sensitivity checks.
Answer: Thank you for this critique. We have conducted an analysis of the percentage of the common contamination-causing genes in single-cell RNA-seq samples: mitochondrial genes (MT-*), ribosomal genes (RPS-*, RPL-*), hemoglobin genes (HBA, HBB), and cell stress genes (FOS, JUN, HSPA1A). None of the genes in this group exhibited an absolute percentage over 1%. The table with the results of this analysis can be viewed in Supplementary Tables (Table S3. QC metrics, identification of tumor cells, cell counts, and sensitivity checks).
3.6 Statistical reporting & robustness
Unresolved. No quantitative sensitivity analyses are shown for DEG thresholds or ligand–receptor inferences; effect sizes and adjusted p‑values need to be consistently reported.
Answer: Thank you for this critique.
Regarding the sensitivity analysis for the The CellChat method, which we use to make ligand-receptor predictions in this study, it includes by default the procedure of statistical significance evaluation, based on the permutation test. For each interaction, the probability is calculated according to the law of action masses, based on averaged level of ligand and receptor expression between cell groups. The significance of interactions is calculated according to random permutations of cell labels, followed by the re-calculation of interaction probability (see the Methods section and Fig. 1c in the original research article CellChat: Jin, S., Guerrero-Juarez, C.F., Zhang, L. et al. Inference and analysis of cell-cell communication using CellChat. Nat Commun 12, 1088 (2021). https://doi.org/10.1038/s41467-021-21246-9). Additionally, the creators of CellChat comment on this subject: “The statistical test is used to find the interactions that are over-represented in certain cell groups. Thus we think it is sufficient. In addition, since we only permute the cell labels 100 times, it is not suitable for further multiple testing. Finally, based on a recent benchmarking study (one paper published in Genome Biol 2023), our method ranks on the top.” (https://github.com/sqjin/CellChat/issues/706)
For DEG analysis and cell interaction analysis, the effect sizes in the form of Log fold change and the adjusted p-values are listed in the respective Methods sections (Lines 466-467, 479).
3.7 Reproducibility and data access
Unresolved. ‘On‑request’ access is not equivalent to public deposition. Processed matrices, metadata, and scripts should be archived with a DOI if de‑identification permits.
Answer: Thank you for this inquiry. The processed dataset, cluster annotations, and analysis scripts have been added to the Zenodo repository under the accession number 17349024 (http://doi.org/10.5281/zenodo.17349024).
4) Major Comments
4.1 Protein‑level validation for key axes
At minimum, provide IHC/IF for selected ligand–receptor pairs central to the claims (e.g., tumor MIF with CD74‑positive myeloid/B‑cell compartments). If archival tissue is insufficient, explicitly state limitations and present the section as future work.
Answer: Thank you for this critique. As we stated in the previous review round, we were not able to perform IHC/IF or spatial transcriptomics on tissue from the same sample, due to the rarity of the tissue. We have modified the limitation section (Lines 391-392) and added a section on future work that should be performed (Lines 396-400) to address this issue.
4.2 Quantitative robustness of inferences
Report how varying DEG thresholds (e.g., min.pct, logFC) and CellChat parameters alters top interactions; include permutation/bootstrapping to estimate false‑positive rates.
Answer:
Thank you for this critique.
We have added a Jaccard index heatmap detailing how the minimum cell percentage cutoff and the log fold change cutoff influence the top interactions to the Supplementary Tables (Table S3. QC metrics, identification of tumor cells, cell counts, and sensitivity checks). Below we will explain our reasoning for choosing the thresholds we used in our study.
The CellChat method by default includes the procedure of evaluation of statistical significance of interactions based on the Permutation Test. For each interaction the probability is calculated according to the law of mass action, based on the average level of expression of ligands and receptors in the analysed cell groups. The significance of interactions is defined through random permutation of cell group labels with consequent recalculation of interaction probabilities (see the Methods section and Fig. 1c in the original research article CellChat: Jin, S., Guerrero-Juarez, C.F., Zhang, L. et al. Inference and analysis of cell-cell communication using CellChat. Nat Commun 12, 1088 (2021). https://doi.org/10.1038/s41467-021-21246-9). Therefore, the levels of false positive results are regulated by the built-in Permutation Test.
The default settings of CellChat function identifyOverExpressedGenes are as follows: thresh.pc = 0, thresh.fc = 0, thresh.p = 0.05. We have examined experimental articles that use CellChat for cell-cell communication analysis, and they typically use these default settings or do not report the DEG thresholds:
- Chen, Lx., Zeng, Sj., Liu, Xd. et al. Cell–cell communications shape tumor microenvironment and predict clinical outcomes in clear cell renal carcinoma. J Transl Med 21, 113 (2023).
- Li, L.X.; Zhang, X.; Zhang, H.; Agborbesong, E.; Zhou, J.X.; Calvet, J.P.; Li, X. Single-Cell and CellChat Resolution Identifies Collecting Duct Cell Subsets and Their Communications with Adjacent Cells in PKD Kidneys. Cells 2023, 12, 45.
- Fang Z, Tian Y, Sui C, Guo Y, Hu X, Lai Y, Liao Z, Li J, Feng G, Jin L and Qian K (2022) Single-Cell Transcriptomics of Proliferative Phase Endometrium: Systems Analysis of Cell–Cell Communication Network Using CellChat. Front. Cell Dev. Biol. 10:919731.
- Chen, W., Zeng, S., Zhong, J. et al. Mapping immune cell dynamics and macrophage plasticity in breast cancer tumor microenvironment through single-cell analysis. Discov Onc 16, 625 (2025).
Therefore, in our study we chose to use the default DEG settings, with slightly elevated minimal cell percentage (thresh.pc = 0.1, thresh.fc = 0, thresh.p = 0.05) so as to take a broad range of significant interactions into account. We acknowledge that these results are hypothetical and IHC confirmations of the discussed interactions are needed.
4.3 Primary vs Supplementary integration of CNV results
Move one CNV heatmap to the main figures and summarise malignant‑fraction per cluster in the Results; this is foundational to all downstream analyses.
Answer: Thank you for this suggestion. We have added the CNV heatmap to Figure 2A. We have also expanded the summary of the malignant cell fraction per cluster in the Results section (Lines 138-141).
4.4 Clarify tumor vs TME classification errors
Discuss possible false positives/negatives in CNV calling (e.g., stromal aneuploidy, ambient RNA), and provide cluster-level confidence metrics or QC flags.
Answer: Thank you for this suggestion. The default program package does not include a detailed analysis of false-positive result statistics and is positioned as a method for CNA (Copy Number Alterations) detection and consequent identification of tumor cells, and for the use of variational algorithm to detect the clonal copy number substructure of tumors from scRNA-seq data. To test for the possible false positives/negatives in CNV calling we have conducted repeats of CNA analysis using the SCEVAN package with the provided ‘normal’ reference (represented by selected annotated T cells) and in the absence of the provided ‘normal’ reference (automatic selection of normal-like cells). The convergence rate between data in these annotations was 97.16289% using Jaccard index, which constitutes a high level of convergence for results of tumor population detection regardless of reference selection. Considering this data, the likelihood of false positives/negatives in CNV calling can be considered minimal. Additional cluster-level confidence metrics would not be informative, due to SCEVAN integrating CNV signals across the entire genome instead of relying on localized/cluster-level statistical thresholds.
4.5 Reproducibility package
Deposit de‑identified gene×cell matrices, cluster annotations, and analysis scripts/notebooks in a public repository (Zenodo/Figshare/GEO) with an accession number cited in Data Availability.
Answer: Thank you for this suggestion. The processed dataset, cluster annotations, and analysis scripts have been added to the Zenodo repository under the accession number 17349024 (http://doi.org/10.5281/zenodo.17349024).
4.6 Tone and claims
Ensure all translational statements are hypothesis‑generating; avoid implying clinical actionability without orthogonal validation and replication.
Answer: Thank you for this critique. We have checked the manuscript again, and made changes that ensure that the translational statements are hypothesis-generating. We have emphasized that while we can make hypotheses based on cell interaction data, these hypotheses are not clinically actionable without larger cohorts, protein verification, and in vitro and in vivo experiments (Lines 26-28, 79, 292-293, 296, 299, 300-301, 302, 305, 310, 315, 320-321, 372-375, 378-385, 391-392, 396-400, 403-405).
5) Minor Comments
– Provide exact cell counts per cluster and their percentage of total; include a compact QC table (median nFeature/nCount per cluster; doublet rate; % filtered).
Answer: Thank you for this inquiry. The cell counts per cluster and their percentage of total, and a QC table per cell cluster are provided in the Supplementary Tables (Table S3. QC metrics, identification of tumor cells, cell counts, and sensitivity checks).
– Standardise gene symbol formatting (italics where required) and check for residual typos; unify British/American spelling across the manuscript.
Answer: Thank you for this comment. We have standardised gene symbol formatting, corrected leftover typos, and unified British/American spelling across the manuscript.
– Specify the statistical test and multiple‑testing correction in every figure/table caption; report adjusted p‑values in Supplementary tables.
Answer: Thank you for this comment. The statistical test names, p-value thresholds, and multiple-testing corrections are available for all 4 Figures in their respective figure legends (Lines 115-121, 157-162, 166-168, 240-242, 334-335).
– Add/confirm scale bars on all histology images and ensure UMAP/dot‑plot font sizes are legible at print scale.
Answer: Thank you for this critique. We increased the font sizes in UMAP/dotplots of Figures 1-4 to make sure the writing is eligible at print scale. We have also confirmed the correctness of the scale bars on histology images.
– State whether any batch effects (lanes/libraries) were present and how integration (e.g., SCTransform) handled them.
Answer: Thank you for this comment. Batch effects were absent due to the presence of a single sample in the study.
– Clarify doublet removal strategy and how potential tumor–stromal doublets were adjudicated.
Answer: Thank you for this inquiry. The doublet/duplicate detection and removal strategy is outlined in Methods, in Lines 456-459.
– Define all abbreviations at first use (e.g., TAM, CAF, Treg); keep terminology consistent throughout.
Answer: Thank you for this correction. We have defined all abbreviations at first use and made the terminology consistent throughout.
6) Recommendation
Major Revision. Version2 is improved (clearer language, better QC/legend reporting, supplementary CNV visuals), but the study remains constrained by single‑case scope, absence of protein‑level validation, limited robustness analyses, and lack of public reproducibility materials. Addressing the above major points would bring the manuscript closer to a publishable standard.
Answer: Thank you so much for the helpful critique. We hope that the added changes to the language of the study, the added robustness analyses, and the deposited public reproducibility materials made our study better and easier to understand.
Round 3
Reviewer 2 Report
Comments and Suggestions for Authors
1) Summary
The authors present a single-patient scRNA-seq case study of pleomorphic rhabdomyosarcoma (pRMS), annotating tumor and TME compartments, inferring malignant fractions via CNV-guided calls, and analyzing cell–cell communication with CellChat. The v3 revision integrates a representative CNV heatmap into the main figures, clarifies exhaustion and myeloid checkpoint marker panels, tones down translational claims, and provides a public Zenodo deposition of processed data and code.
2) Evaluation of Revisions (Point-by-Point)
Therapeutic over-interpretation — Partially Resolved.
Language in the Abstract/Discussion now frames implications as hypothesis-level rather than actionable, which is appropriate for a single case; however, a few phrases still imply therapeutic promise. Keep all such statements explicitly speculative.
Single-patient scope & external validity — Unresolved.
The study still reports one specimen without orthogonal validation or an independent case. The limitation is acknowledged; please reiterate this prominently in the Abstract and Conclusions.
Marker panels (T-cell exhaustion; myeloid checkpoints) — Resolved.
The CD8+ exhaustion signature is corrected to LAG3/HAVCR2/EOMES/PDCD1, and the term “myeloid checkpoint-related” replaces “myeloid exhaustion,” with consistent Figure 3 text/legend. (v1 erroneously included CD74 in the exhaustion panel; v3 corrects this.)
CNV-guided tumor calling (SCEVAN) & reporting — Resolved.
A representative CNV heatmap is now in the main figures, and per-cluster malignant fractions are reported in the Results (Lines ~137–143).
QC / ambient RNA handling — Partially Resolved.
The authors add quantitative checks for contamination-prone gene families (generally <1% absolute). Please also report doublet identification methods, thresholds, estimated doublet rate, and the number of cells removed.
Statistical reporting & robustness — Partially Resolved.
A Jaccard-index sensitivity analysis for DEG thresholds is added in Supplementary materials, and CellChat’s built-in permutation tests are cited. Please report the number of permutations, per-interaction effect sizes/probabilities, adjusted p-values (FDR), and how multiple testing was controlled for LR inferences (main text or a compact table).
Reproducibility & data access — Resolved.
Processed matrices, annotations, and scripts are deposited on Zenodo (17349024) and cited in the Data Availability statement.
3) Major Comments (Remaining Issues)
- Strengthen the limitations. Explicitly state in the Abstract and Conclusions that this is a single-patient transcriptomic case study without protein-level or spatial validation, and no clinical actionability is claimed.
- Protein-level/orthogonal support. If archival tissue is unavailable, acknowledge it prominently and, at minimum, provide orthogonal evidence via re-analysis of public pRMS/soft-tissue sarcoma datasets (bulk RNA/IHC repositories) to assess MIF–CD74, APP, PTN, CXCL12 axes across tumor–immune compartments. Keep all therapeutic language hypothesis-generating.
- CellChat inferences. Report # permutations, interaction probabilities/effect sizes, FDR-adjusted p-values per edge; and include a compact table of top ligand–receptor pairs with these statistics. Ensure the Jaccard heatmap referenced in the response appears in Supplementary and summarize its implications in the main text.
- Doublets and ambient RNA. Explicitly list which doublet-detection tools were used (e.g., DoubletFinder/scDblFinder/Scrublet), thresholds, estimated rates, and cells removed per cluster; add a brief SoupX/DecontX summary or rationale for not applying them. Provide the requested compact QC table (per-cluster nFeature_RNA, nCount_RNA, %mt; doublet rate; % filtered).
- CNV calling robustness. Along with Figure 2A, include a concise per-cluster malignant-fraction table in the main text (not only Supplementary), and briefly justify the <30% aneuploidy rule for “normal” clusters. Consider reporting concordance when varying the normal reference and SCEVAN parameters (you already provide convergence rationale in the response; summarize it in Results/Methods).
- Terminology consistency. Ensure uniform use of “myeloid checkpoint-related” (not “exhaustion”) and correct gene symbols (CSF1R, not CSFR1). Confirm consistency in all figure legends (Figure 3E) and the main text.
4) Minor Comments
- Provide exact cell counts per cluster in the Results (also present in figure legends) and ensure they match Supplemental tables.
- Standardize British/American spelling (e.g., signalling/signaling) and hyphenation; do a careful proofread for minor typos.
- Clarify that CD24/CDH2 are discussed as interaction-related or adhesion/immune-modulatory markers rather than “exhaustion” genes; align wording with v3.
- Explicitly list DEG thresholds for each comparison (log2FC, min.pct, adjusted p-value) in the main text to mirror Methods; report effect sizes.
- If journal style allows, cite the Zenodo DOI also in the Abstract’s data/resource statement.
5) Conclusion
The authors have substantially improved the manuscript in v3 (marker panels, CNV visualization/reporting, data deposition, and toned-down claims). Nonetheless, given the single-patient scope without orthogonal validation and the remaining gaps in quantitative robustness/QC reporting, I recommend Major Revision. With the clarifications and additions above, the manuscript could become suitable for publication.
Author Response
2) Evaluation of Revisions (Point-by-Point)
2.1 Therapeutic over-interpretation — Partially Resolved.
Language in the Abstract/Discussion now frames implications as hypothesis-level rather than actionable, which is appropriate for a single case; however, a few phrases still imply therapeutic promise. Keep all such statements explicitly speculative.
Answer: Thank you for this critique. Line 26-28 in the Abstract have been changed, and the Conclusions paragraph (Lines 432-436) has been changed and shortened in order to avoid therapeutic over-interpretation.
2.2 Single-patient scope & external validity — Unresolved.
The study still reports one specimen without orthogonal validation or an independent case. The limitation is acknowledged; please reiterate this prominently in the Abstract and Conclusions.
Answer: Thank you for this critique. The phrase “this is a single-patient transcriptomic case study without protein-level or spatial validation, and no clinical actionability is claimed” has been prominently added to the Conclusions (Lines 434-436). Lines 26-28 in the Abstract have also been further altered to reflect this sentiment.
2.3 Marker panels (T-cell exhaustion; myeloid checkpoints) — Resolved.
The CD8+ exhaustion signature is corrected to LAG3/HAVCR2/EOMES/PDCD1, and the term “myeloid checkpoint-related” replaces “myeloid exhaustion,” with consistent Figure 3 text/legend. (v1 erroneously included CD74 in the exhaustion panel; v3 corrects this.)
Answer: Thank you for your critique. It has helped us improve this work.
2.4 CNV-guided tumor calling (SCEVAN) & reporting — Resolved.
A representative CNV heatmap is now in the main figures, and per-cluster malignant fractions are reported in the Results (Lines ~137–143).
Answer: Thank you for your critique. It has helped us improve this work.
2.5 QC / ambient RNA handling — Partially Resolved.
The authors add quantitative checks for contamination-prone gene families (generally <1% absolute). Please also report doublet identification methods, thresholds, estimated doublet rate, and the number of cells removed.
Answer: Thank you for this critique.
The doublet/duplicate detection and removal strategies and estimated doublet rates are outlined in Methods (Lines 496-499). The per-cluster nFeature_RNA, nCount_RNA, and percent_mt are provided in Table S3. QC metrics, identification of tumor cells, cell counts, sensitivity checks. The percent of removed cells and the doublet/duplicate rate is provided in Line 126-127 of Results. We have also added the percent of removed cells to the Method section (Line 494-495), so both the doublet rate and the filtered cell percentage is duplicated there.
We have added our rationale for not using SoupX to the Method section: “The percentage of expressed genes that are considered as widely known contaminants in single-cell RNA-seq samples: mitochondrial genes (MT-*, where * represents a specific gene identifier), ribosomal genes (RPS-*, RPL-*, where * represents a specific gene identifier), hemoglobin genes (HBA, HBB), and cell stress genes (FOS, JUN, HSPA1A) was analyzed. None of the genes in this group exhibited an absolute percentage over 1%. A decision was made not to use SoupX to remove RNA contamination, since the data did not contain elevated values of contaminating RNAs and avoiding the possible effects of expression matrix correction on data interpretation was preferable.” (Lines 486-494).
2.6 Statistical reporting & robustness — Partially Resolved.
A Jaccard-index sensitivity analysis for DEG thresholds is added in Supplementary materials, and CellChat’s built-in permutation tests are cited. Please report the number of permutations, per-interaction effect sizes/probabilities, adjusted p-values (FDR), and how multiple testing was controlled for LR inferences (main text or a compact table).
Answer: Thank you for this critique.
The CellChat developers do not recommend multiple testing, since the p-value is calculated using a formula that takes into account permutation values ​​and outputs only the p-value and interaction probabilities for ligand-receptor pathways. The default number of permutations in CellChat in 100. We have added this information to the Methods section (Lines 531-533).
We have added all the calculated probabilities and p-values for all analysed pathways for our sample to the Table S4. CellChat probability full data. The Jaccard index heatmap, detailing how the minimum cell percentage cutoff and the log fold change cutoff influence the top interactions, as well as the top pathway comparison for different minimum cell percentage cutoffs and log fold change cutoffs can be found in the Supplementary Tables (Table S3. QC metrics, identification of tumor cells, cell counts, sensitivity checks). We have also added Lines 533-538 to direct our readers to these supplementary files in the Methods section.
2.7 Reproducibility & data access — Resolved.
Processed matrices, annotations, and scripts are deposited on Zenodo (17349024) and cited in the Data Availability statement.
Answer: Thank you for your critique. It has helped us improve this work.
3) Major Comments (Remaining Issues)
3.1 Strengthen the limitations. Explicitly state in the Abstract and Conclusions that this is a single-patient transcriptomic case study without protein-level or spatial validation, and no clinical actionability is claimed.
Answer: Thank you for this critique. The phrase “this is a single-patient transcriptomic case study without protein-level or spatial validation, and no clinical actionability is claimed” has been added to the Conclusions (Lines 434-436). Line 26-28 in the abstract has been changed to reflect the same sentiment. Additionally, the fact that this study is a single-case report is evident from the article category it has been submitted to (Case Report), which is explicitly stated in Line 1.
3.2 Protein-level/orthogonal support. If archival tissue is unavailable, acknowledge it prominently and, at minimum, provide orthogonal evidence via re-analysis of public pRMS/soft-tissue sarcoma datasets (bulk RNA/IHC repositories) to assess MIF–CD74, APP, PTN, CXCL12 axes across tumor–immune compartments. Keep all therapeutic language hypothesis-generating.
Answer: Thank you for this critique.
We have added Lines 409-410 to the Discussion, in which we prominently acknowledge the lack of archival tissue to perform IHC.
Bulk transcriptomic data for pRMS exists (Beird et al, 2023). However, we do not see how we can verify the transcriptional expression of MIF and APP in tumor populations and the expression in CD74 in specific immune populations with bulk data. MIF and APP expression has been previously described for a wide range of cells, from tumor cells to immune cells and fibroblasts.
In the Discussion (Lines 410-416) we have added a discussion of the single-cell study (Tessaro et al, 2022) that demonstrated the presence MIF-CD74 and APP-CD74 tumor-immune interactions on the transcriptional level in the mouse model of UPS, a sarcoma type closely related to pRMS. This study also shows the expression of MIF on the protein level in human sarcoma cell lines and the therapeutic in vitro effect of MIF-CD74 tumor-immune axis silencing.
Finally, we would like to stress that our article is a case report (the article category is listed in Line 1), which are typically concise, descriptive, and feature a limited number of methods used to characterize a single clinical case.
3.3 CellChat inferences. Report # permutations, interaction probabilities/effect sizes, FDR-adjusted p-values per edge; and include a compact table of top ligand–receptor pairs with these statistics. Ensure the Jaccard heatmap referenced in the response appears in Supplementary and summarize its implications in the main text.
Answer: Thank you for this critique.
The CellChat developers do not recommend multiple testing, since the p-value is calculated using a formula that takes into account permutation values ​​and outputs only the p-value and interaction probabilities for ligand-receptor pathways. The default number of permutations in CellChat in 100. We have added this information to the Methods section (Lines 531-533).
We have added all the calculated probabilities and p-values for all analysed pathways for our sample to the Table S4: CellChat probability full data. The Jaccard index heatmap, detailing how the minimum cell percentage cutoff and the log fold change cutoff influence the top interactions, as well as the top pathway comparison for different minimum cell percentage cutoffs and log fold change cutoffs can be found in the Supplementary Tables (Table S3. QC metrics, identification of tumor cells, cell counts, sensitivity checks). We have also added Lines 533-538 to direct our readers to these supplementary files in the Methods section.
3.4 Doublets and ambient RNA. Explicitly list which doublet-detection tools were used (e.g., DoubletFinder/scDblFinder/Scrublet), thresholds, estimated rates, and cells removed per cluster; add a brief SoupX/DecontX summary or rationale for not applying them. Provide the requested compact QC table (per-cluster nFeature_RNA, nCount_RNA, %mt; doublet rate; % filtered).
Answer: Thank you for this critique.
The doublet/duplicate detection and removal strategy and estimated doublet rates are outlined in Methods (Lines 496-499). The per-cluster nFeature_RNA, nCount_RNA, and percent_mt are provided in Table S3. QC metrics, identification of tumor cells, cell counts, sensitivity checks. The percent of removed cells and the doublet/duplicate rate is provided in Line 126-127 of Results. We have also added the percent of removed cells to the Method section (Line 494-495), so both the doublet rate and the filtered cell percentage is duplicated there.
We have added our rationale for not using SoupX to the Method section: “The percentage of expressed genes that are considered as widely known contaminants in single-cell RNA-seq samples: mitochondrial genes (MT-*, where * represents a specific gene identifier), ribosomal genes (RPS-*, RPL-*, where * represents a specific gene identifier), hemoglobin genes (HBA, HBB), and cell stress genes (FOS, JUN, HSPA1A) was analyzed. None of the genes in this group exhibited an absolute percentage over 1%. A decision was made not to use SoupX to remove RNA contamination, since the data did not contain elevated values of contaminating RNAs and avoiding the possible effects of expression matrix correction on data interpretation was preferable.” (Lines 486-494).
3.5 CNV calling robustness. Along with Figure 2A, include a concise per-cluster malignant-fraction table in the main text (not only Supplementary), and briefly justify the <30% aneuploidy rule for “normal” clusters. Consider reporting concordance when varying the normal reference and SCEVAN parameters (you already provide convergence rationale in the response; summarize it in Results/Methods).
Answer: Thank you for this critique.
We have added pie diagrams of per-cluster malignant cell fractions to Figure 2 (Fig. 2B) and the respective changes to the Figure 2 legend (Line 160-161). We have added an explanation for our rationale for using the <30% aneuploidy rule for normal cell clusters (Lines 143-144). We have added the paragraph on convergence rationale to the Methods section (Lines 517-526).
3.6 Terminology consistency. Ensure uniform use of “myeloid checkpoint-related” (not “exhaustion”) and correct gene symbols (CSF1R, not CSFR1). Confirm consistency in all figure legends (Figure 3E) and the main text.
Answer: Thank you for this critique. We have ensured uniform use of “myeloid checkpoint-related” (not “exhaustion”) to describe the state of macrophages in our case study of pRMS. For this purpose Line 314 has been altered to read “Considering the signs of exhaustion in T cell clusters and the myeloid-checkpoint related genes in macrophage clusters, we focused on the effect that pRMS tumor and immune cell clusters could potentially exert on each other.” We have also checked for the correct spelling of CSF1R (not CSFR1). The consistency in all figure legends (Figure 3E) and the main text is confirmed.
4) Minor Comments
4.1 Provide exact cell counts per cluster in the Results (also present in figure legends) and ensure they match Supplemental tables.
Answer: Thank you for this critique. We have added cell counts per cluster in the Results (Lines 134-137, 177-180, 226-228). They match the numbers presented in the figure legends and provided in the Supplementary tables.
4.2 Standardize British/American spelling (e.g., signalling/signaling) and hyphenation; do a careful proofread for minor typos.
Answer: Thank you for this critique. We have standardised British/American spelling and hyphenation across the manuscript and done a proofread for minor typos.
4.3 Clarify that CD24/CDH2 are discussed as interaction-related or adhesion/immune-modulatory markers rather than “exhaustion” genes; align wording with v3.
Answer: Thank you for this critique. CD24 and CDH2 are only mentioned once in the manuscript in Line 156-157 and are described as having “interaction with the immune cells” (not as exhaustion genes) as per the final V3 of the manuscript.
4.4 Explicitly list DEG thresholds for each comparison (log2FC, min.pct, adjusted p-value) in the main text to mirror Methods; report effect sizes.
Answer: Thank you for this critique. We have added the DEG thresholds for each comparison (Log2 Fold Change, pct., adj. p-value) throughout the Results section.
4.5 If journal style allows, cite the Zenodo DOI also in the Abstract’s data/resource statement.
Answer: Thank you for this critique. The IJMS Instructions for Authors do not mention the data/resource statement section of the Abstract. Therefore, we added the statement with the Zenodo DOI for our dataset to the beginning of the Results section (Lines 127-129).
5) Conclusion
The authors have substantially improved the manuscript in v3 (marker panels, CNV visualization/reporting, data deposition, and toned-down claims). Nonetheless, given the single-patient scope without orthogonal validation and the remaining gaps in quantitative robustness/QC reporting, I recommend Major Revision. With the clarifications and additions above, the manuscript could become suitable for publication.
Answer: Thank you so much for the helpful critique. We hope that the added changes to the quantitative robustness/QC reporting, language changes, and other clarifications made our study better and easier to understand.
Round 4
Reviewer 2 Report
Comments and Suggestions for Authors
1) Summary & scope
The manuscript reports a single-patient single-cell RNA-sequencing (scRNA-seq) case study of pleomorphic rhabdomyosarcoma (pRMS). The analysis includes QC and clustering, malignant versus non-malignant cell identification using CNV inference, tumor–microenvironment transcriptional contrasts, and putative ligand–receptor communication with CellChat. Public availability of data/code is stated. Compared with prior rounds, v4 appropriately tempers claims and adds an explicit limitations statement, which is appropriate for a single-patient case report.
2) Major strengths (as revised)
- Reproducibility & access. Processed matrices/annotations/scripts are publicly accessible and clearly cited.
- Transparent QC and ambient-RNA rationale. Doublet detection and contamination checks are reported; the decision not to apply ambient-RNA correction is explicitly justified.
- Explicit DEG thresholds. Statistical framework (e.g., MAST), effect-size and FDR criteria are reported consistently.
- CNV robustness. A sensitivity check indicates stable malignant fraction estimates regardless of reference selection.
- Cell–cell communication statistics. Permutation-based significance is described; per-edge probability/p-value outputs are referenced in the supplement.
- Tempered claims & context. Conclusions emphasize hypothesis generation and the absence of orthogonal validation, aligned with the case-report scope.
3) Points requiring attention before acceptance
- Numerical consistency for cell counts and filtering
Ensure the total cells, filtered cells, and percentages are consistent across Results, Methods, figures, and Supplementary materials. Where you report a percentage, include the numerator/denominator so readers can reconcile the arithmetic.
- CellChat multiplicity statement
Add one clarifying sentence in Methods that p-values follow CellChat’s built-in permutation framework (label shuffling) and that no additional FDR layer is applied beyond CellChat’s defaults. This prevents confusion about multiple testing given the large number of ligand–receptor edges.
- Terminology consistency (myeloid/“checkpoint-related”)
Do a final pass to harmonize terminology and spelling across text and all figure legends (e.g., CSF1R; avoid labeling CD24/CDH2 as “exhaustion” markers; prefer “checkpoint-related”/“immunomodulatory” phrasing where applicable).
- Per-cluster counts cross-check
Verify that per-cluster cell counts in the main text exactly match figure labels and Supplement tables. Consider pointing to the exact Supplement table/sheet name to aid traceability.
- Limitations statement placement
Keep the tempered, hypothesis-generating language in both Abstract and Conclusions. Explicitly reiterate the single-patient scope and the need for orthogonal validation (e.g., IHC/ISH or independent cohorts).
4) Minor suggestions
- At the first mention of the public repository in Results, append “(see Data Availability).”
- For full computational reproducibility, either list the clustering/UMAP parameters (e.g., resolution, n.neighbors, min.dist) or point to the exact script and parameter file in the repository.
- Consider moving one sentence of the ambient-RNA rationale into Results (with a pointer back to Methods), so non-methods-focused readers understand why ambient correction was not applied.
5) Comparative assessment across versions
From v1 to v4, the manuscript (i) adds public data/code access; (ii) consolidates DEG thresholds and FDR reporting; (iii) clarifies CellChat’s significance procedure; (iv) introduces a CNV robustness check; and (v) standardizes terminology while tempering claims to match a case-report design.
6) Recommendation
Recommendation: Minor Revision.
Rationale: v4 substantially improves methodological transparency, data availability, and interpretative restraint. With (i) cell-count arithmetic reconciled across sections, (ii) a one-sentence clarification on CellChat multiplicity, and (iii) final terminology/label checks, the manuscript will meet publication standards for a well-documented, hypothesis-generating scRNA-seq case report.
Author Response
3) Points requiring attention before acceptance
1. Numerical consistency for cell counts and filtering
Ensure the total cells, filtered cells, and percentages are consistent across Results, Methods, figures, and Supplementary materials. Where you report a percentage, include the numerator/denominator so readers can reconcile the arithmetic.
Answer:Thank you for this critique. We have checked again to make sure that the total cells, filtered cells, and percentages are reported consistently across the Results, Methods, figures, and Supplementary materials. We have added the numerator/denominator ratios to the cell percentages in the Results and Methods sections (Lines 127, 128, 147-149, 155, 171, 306, 504-505, 509).
2. CellChat multiplicity statement
Add one clarifying sentence in Methods that p-values follow CellChat’s built-in permutation framework (label shuffling) and that no additional FDR layer is applied beyond CellChat’s defaults. This prevents confusion about multiple testing given the large number of ligand–receptor edges.
Answer: Thank you for this critique. The sentence “P-values follow CellChat’s built-in permutation framework (label shuffling), and no additional FDR layer is applied beyond CellChat’s defaults.” has been added to the Methods section (Lines 550-552).
3. Terminology consistency (myeloid/“checkpoint-related”)
Do a final pass to harmonize terminology and spelling across text and all figure legends (e.g., CSF1R; avoid labeling CD24/CDH2 as “exhaustion” markers; prefer “checkpoint-related”/“immunomodulatory” phrasing where applicable).
Answer: Thank you for this critique. We have checked the text to ensure the correct spelling of CSF1R (not CSFR1). The CD24/CDH2 are not referred to as exhaustion markers anywhere in the manuscript. The consistency of these terms across all figures, figure legends, and the main text is confirmed. The phrases “checkpoint-related”/“immunomodulatory” are preferred where applicable.
4. Per-cluster counts cross-check
Verify that per-cluster cell counts in the main text exactly match figure labels and Supplement tables. Consider pointing to the exact Supplement table/sheet name to aid traceability.
Answer: Thank you for this critique. We have verified that per-cluster cell counts in the main text exactly match figure labels and Supplement tables. The exact Supplement table/sheet names have been added throughout the manuscript (Example: Figure 2A, B; Table S3, Sheet2. Validation of tumor vs TME)(Lines 150-151) to aid traceability of data.
5. Limitations statement placement
Keep the tempered, hypothesis-generating language in both Abstract and Conclusions. Explicitly reiterate the single-patient scope and the need for orthogonal validation (e.g., IHC/ISH or independent cohorts).
Answer: Thank you for this critique. As of the final version of the manuscript, we preserve the tempered, hypothesis-generating language in both Abstract and Conclusions. The Lines 26-28 in the Abstract, Lines 415-417, 426-440 in the Discussion, and Lines 443-445 in the Conclusions reiterate the the single-patient scope and the need for orthogonal validation (e.g., IHC/ISH or independent cohorts).
4) Minor suggestions
- At the first mention of the public repository in Results, append “(see Data Availability).”
Answer: Thank you for this critique. The phrase “(Data Availability)” has been added to the Results section (Line 131-132), at the first mention of the public repository.
- For full computational reproducibility, either list the clustering/UMAP parameters (e.g., resolution, n.neighbors, min.dist) or point to the exact script and parameter file in the repository.
Answer: Thank you for this critique. The sentence “Clustering/UMAP parameters (resolution, n.neighbors) for each figure are described in analysis script pRMS analysis.R published in Zenodo repository (Line 41-44,145-151,188-193; Data Availability).” (Lines 512-514).
- Consider moving one sentence of the ambient-RNA rationale into Results (with a pointer back to Methods), so non-methods-focused readers understand why ambient correction was not applied.
Answer: Thank you for this critique. We have added the sentence “For this dataset, SoupX was not used to remove RNA contamination, since the data did not contain elevated values of contaminating RNAs (section 4.6 in Methods contains the full ambient-RNA rationale explanation).” to the Results section (Lines 128-130).
5) Comparative assessment across versions
From v1 to v4, the manuscript (i) adds public data/code access; (ii) consolidates DEG thresholds and FDR reporting; (iii) clarifies CellChat’s significance procedure; (iv) introduces a CNV robustness check; and (v) standardizes terminology while tempering claims to match a case-report design.
Answer: Thank you for your critique. It has helped us improve this work.
6) Recommendation
Recommendation: Minor Revision.
Rationale: v4 substantially improves methodological transparency, data availability, and interpretative restraint. With (i) cell-count arithmetic reconciled across sections, (ii) a one-sentence clarification on CellChat multiplicity, and (iii) final terminology/label checks, the manuscript will meet publication standards for a well-documented, hypothesis-generating scRNA-seq case report.
Answer: Thank you so much for the helpful critique. We hope that the reconciliation of the cell-count arithmetic across sections, the added clarification on CellChat multiplicity, and the final terminology/label checks made our scRNA-seq case report better and easier to understand.